# TWO STAGES DOMAIN INVARIANT REPRESENTATION LEARNERS SOLVE THE LARGE CO-VARIATE SHIFT IN UNSUPERVISED DOMAIN ADAPTATION WITH TWO DIMENSIONAL DATA DOMAINS

## ABSTRACT

Recent developments in the unsupervised domain adaptation (UDA) enable the unsupervised machine learning (ML) prediction for target data, thus this will accelerate real world applications with ML models such as image recognition tasks in self-driving. Researchers have reported the UDA techniques are not working well under large co-variate shift problems where e.g. supervised source data consists of handwritten digits data in monotone color and unsupervised target data colored digits data from the street view. Thus there is a need for a method to resolve co-variate shift and transfer source labelling rules under this dynamics. We perform two stages domain invariant representation learning to bridge the gap between source and target with semantic intermediate data (unsupervised). The proposed method can learn domain invariant features simultaneously between source and intermediate also intermediate and target. Finally this achieves good domain invariant representation between source and target plus task discriminability owing to source labels. This induction for the gradient descent search greatly eases learning convergence in terms of classification performance for target data even when large co-variate shift. We also derive a theorem for measuring the gap between trained models and unsupervised target labelling rules, which is necessary for the free parameters optimization. Finally we demonstrate that proposing method is superiority to previous UDA methods using 4 representative ML classification datasets including 38 UDA tasks. Our experiment will be a basis for challenging UDA problems with large co-variate shift.

## 1 INTRODUCTION

These days UDA is attracting attentions from researchers and engineers in ML projects, since it can automatically correct difference in the marginal distributions of the training source data (supervised) and test target data (unsupervised) and learn the better model based on labeling rule from source data. Especially domain invariant representation learning methods including the domain adversarial training of neural networks (DANNs) (Ganin et al., 2017), the correlation alignment for deep domain adaptation (Deep CoRALs) (Sun & Saenko, 2016), and the deep adaptation networks (DANs) (Long et al., 2015) have achieved performance improvements in a variety of ML tasks for instance digits image recognition and the human activity recognition (HAR) with accelerometer and gyroscope (Wilson et al., 2020). There are emerging projects in a fairly business-like setting with UDA, and demonstrated a certain level of success. For instance image semantic segmentation task in self-driving where different whether conditions data undermines ML models performance due to that distribution gap between training and testing (Liu et al., 2020).

On the other hand, researchers have reported the UDA techniques are not working well under large co-variate shift. The techniques could cope with a small amount of distribution gap, e.g. between monotone color digit images and their colored version without changing background and digit itself in other means, or between simulated binary toy data and their 30 degrees rotated version (Ganin et al., 2017). In contrast to that, reported discrimination performance was not over 25 % and had very high variance using UDA with the convolutional neural networks (CNNs) backbones between

monotone color digit data and colored digit data from street view (Ganin et al., 2017). Same problem was observed between binary toy data and 50-70 degrees rotated version (See Figure 2 that we explain later. We can create much better model for 30 degrees target data using same shallow neural networks backbone (Ganin et al., 2017).). Such UDA problems with large co-variate shift are often in real-world ML tasks. We limit scope of large co-variate shift problem to two dimensional data domains causing co-variate shifts. In this setting two attributes related to data are causing co-variate shifts e.g. from monotone to color and from not on the street to on the street, the details of which are dealt with in 2.2.

In this study we propose two stages domain invariant representation learning to fill this gap. Use intermediate data (unsupervised) between source and target to ensure simultaneous domain invariance between source and intermediate data and invariance between intermediate and final target data, this greatly enhances learning convergence in terms of classification performance for target. We can usually get access to intermediate unsupervised data compared to huge burden for labelling processes involving human labour and expensive measuring equipment. We also demonstrate a theorem that allows unsupervised hyper-parameters optimisation based on the reverse validation (RV) (Zhong et al., 2010). This measures the difference between the target labelling rules and the labelling rules of the model after UDA training without any access to the target supervised labels. The UDA tends to negative transfer with inappropriate free parameters (Wang et al., 2019b), therefore theoretical supported indicator is important.

Experimental results with four datasets confirmed the superiority of the proposed method to previous studies. Datasets for comparison tests with high demand for social implementations include image recognition and accelerometer based HAR, and occupancy detection with energy consumption data measured by smarter maters in general households. This paper contributes to (1) Proposition of UDA strategy as a solution to large co-variate shift, (2) Derivation of free parameters tuning indicator, enables validation for conditional distribution difference without any access for target ground truth labels, (3) Demonstration of experimental superiority after comparison tests with benchmarks using four representative datasets.

## 2 PRELIMINARY

### 2.1 UNSUPERVISED DOMAIN ADAPTATION

Sets of data are comprised of $\mathcal{D}_S = \{(x_i^S, y_i^S)\}_{i=1}^{N_S}$, $\mathcal{D}_T = \{x_i^T\}_{i=1}^{N_T}$, $\mathcal{D}_{T'} = \{x_i^{T'}\}_{i=1}^{N_{T'}}$ (Source domain, intermediate domain, target domain respectively and we denote $N_S, N_T, N_{T'}$ as the sample sizes for source, intermediate and target.), in our setting $\mathcal{D}_T$ is e.g. from (UserA, Summer) when $\mathcal{D}_S$ is from (UserA, Winter) and $\mathcal{D}_{T'}$ is from (UserB, Summer). Then let $P_S(y|x), P_S(x)$ denote the marginal and conditional distribution of source, defined for intermediate and target similarly. They are in the homogeneous domain adaptation assumption, namely they share same sample space but different distribution (Wilson & Cook, 2020). Also this research is in co-variate shift problem, that is generally sharing conditional distribution but different marginal distribution of co-variate (Zhao et al., 2019). The objective is learning $\phi(\cdot) = F \circ C$ to predict ground truth labels for $\mathcal{D}_{T'}$ using three sets of data without access to target ground truth labels, $F$ corresponds to feature extractor with arbitrary neural networks parameter $\theta_f$ and $C$ task classifier with $\theta_c$ to be optimized by gradient descent.

### 2.2 WHAT IS TWO DIMENSIONAL CO-VARIATE SHIFTS

Let's dive into what is intermediate domain $\mathcal{D}_T$, we assume data domain is not one dimensional but is two dimensional e.g. (UserA or UserB, Accelerometer Model A and Accelerometer Model B) in HAR, (UserA or UserB, Winter or Summer) in occupancy detection problem and (Monotone or Color, In the street or not) in image recognition. In this paradigm, $\mathcal{D}_S, \mathcal{D}_T, \mathcal{D}_{T'}$ may be (UserA, Accelerometer Model A), (UserB, Accelerometer Model A), (UserB, Accelerometer Model B) for instance. These two dimensional factors are influencing data distribution (we call this as two dimensional co-variate shifts), energy consumption data differs between users and also seasons for instance. Basically under this two dimensional co-variate shifts problem correcting difference between domains in UDA is quite difficult but more natural case closer to businesses or real world projects. Additionally we assume gathering unsupervised data $\mathcal{D}_T$ is quite easy, so UDA problem

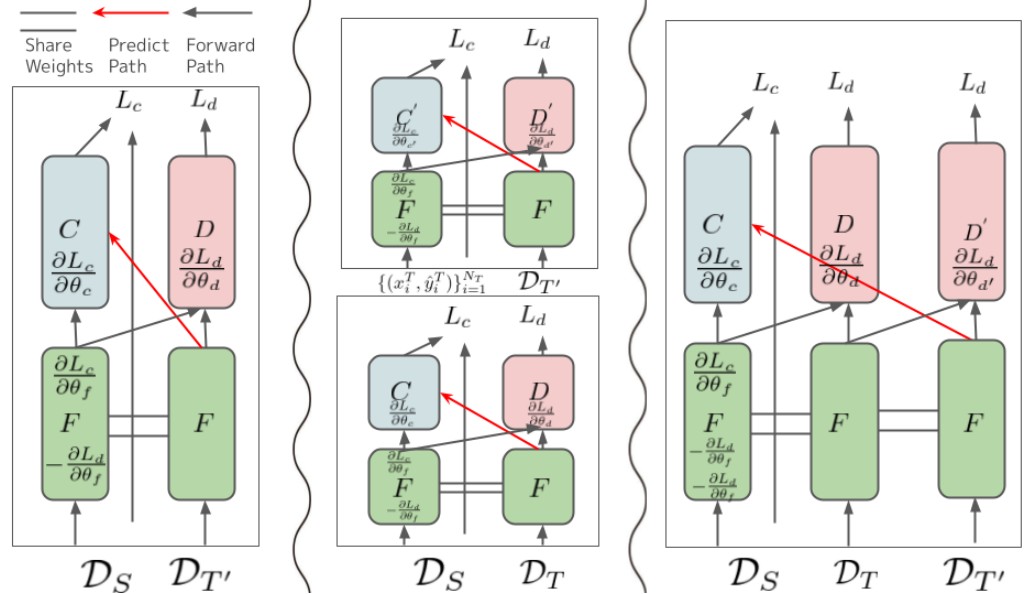

Figure 1: Forward path and backward process of Normal(DANNs), Step-by-step, Ours. The $L_{task}, -L_{domain}$ as $L_c, L_d$ for short.

with three sets $\mathcal{D}_S, \mathcal{D}_T, \mathcal{D}_{T'}$ is natural and there is demand for solving this problem. Two dimensional domains assumption is novel in this field, although uses of an intermediate domain to resolve large co-variate shift are explored in (Lin et al., 2021; Zhang et al., 2019; Oshima et al., 2024) as well. Their experiments showed a synthetic intermediate domain can improve UDA (Lin et al., 2021; Zhang et al., 2019), but limited to computer vision tasks technically or experimentally. Our proposition is not limited to a specific data type. We investigated whether or not each dataset follows this assumption in the Appendix E.

## 3 UDA WHEN TWO DIMENSIONAL CO-VARIATE SHIFTS

### 3.1 LARGE CO-VARIATE SHIFT SOLVER: TWO STAGES DOMAIN INVARIANT LEARNERS

To begin with previous domain invariant learning abstraction, their objective functions and optimization are below (we call this as Normal). Domain invariant learning is parallel learning including the feature extractor and task classifier's learning labelling rule based on $\mathcal{D}_S$ by minimizing $L_{task}$ and the feature extractor's learning domain invariance between $\mathcal{D}_S, \mathcal{D}_{T'}$ by $L_{domain}$ (measurement of distribution gap between source and target, we elaborate later). The constant value $\lambda$ is coefficient of weight to adjust the balance between task classification performance and domain invariant performance. We define $L_{task} = -\sum_{i=1}^{batch\_size} \sum_{j=1}^{num\_class} y_{i,j}^S \log(\hat{y}_{i,j}^S)$ as the cross entropy loss with a predicted probability $\hat{y}_{i,j}^S$ for source input.

$$L = L_{task} + \lambda L_{domain} \tag{1}$$

$$\mathrm{argmin}_{\theta_f, \theta_c} L_{task}, \mathrm{argmin}_{\theta_f, (\theta_c)} L_{domain} \tag{2}$$

This will achieve the feature extractor and task classifier's generalization performance for target data theoretically, since the UDA goal(expectation risk for target discrimination performance) is bounded by marginal distribution difference between source and target (corresponds to $L_{domain}$), empirical risk for source discrimination performance (corresponds to $L_{task}$), conditional distribution gap (under co-variate shift problem this should be small), and so fourth. Let $\mathcal{H}$ be a hypothesis space of VC-dimension $d$, $\hat{\mathcal{D}}_S, \hat{\mathcal{D}}_{T'}$ be empirical sample sets with size $N$ drawn from source joint distribution (features and label) and target marginal distribution (features), $d_i$ be the domain label (binary label identifying source or target, $d_i \in \{0, 1\}$), $\mathbb{E}_{T'}(h), \mathbb{E}_S(h)$ be expectations for hypothesis

$h$ (task classification), $\hat{\mathbb{E}}_S(h)$ be the empirical expectation, $\mathbb{I}[\cdot]$ be the function outputting 1 when true inside the square brackets otherwise outputting 0, and then w.p. at least $1 - \delta$ ($\delta \in (0,1)$), $\forall h \in \mathcal{H}$, the following inequality holds (Zhao et al., 2019; Ben-David et al., 2006).

$$\mathbb{E}_{T'}(h) \leq \hat{\mathbb{E}}_S(h) + \frac{1}{2}d_{\mathcal{H}}(\hat{\mathcal{D}}_S, \hat{\mathcal{D}}_{T'}) + \lambda^* + \mathcal{O}(\sqrt{\frac{d\log N + \log(\frac{1}{\delta})}{N}}) \tag{3}$$

$$h^* = \mathrm{argmin}_{h \in \mathcal{H}}\mathbb{E}_S(h) + \mathbb{E}_{T'}(h), \lambda^* = \mathbb{E}_S(h^*) + \mathbb{E}_{T'}(h^*) \tag{4}$$

$$d_{\mathcal{H}}(\hat{\mathcal{D}}_S, \hat{\mathcal{D}}_{T'}) = 2(1 - \min_{h \in \mathcal{H}}(\frac{1}{N_S}\sum_{i=1}^{N_S}\mathbb{I}[h(x_i^S) \neq d_i^S] + \frac{1}{N_{T'}}\sum_{j=1}^{N_{T'}}\mathbb{I}[h(x_j^{T'}) \neq d_j^{T'}])) \tag{5}$$

Previous studies such as DANNs, CoRALs and DANs have been published as embodiments of equation 1 and 2. The construction of $L_{domain}$ differs across methods. DANNs define it as a loss of domain classification problem, while CoRALs use the distance between covariance matrices of $C$. DANs build it using the multiple kernel variant of maximum mean discrepancy (MK-MMD (Gretton et al., 2012)) calculated on $F$ and $C$, and Deep Joint Distribution Optimal Transportation (DeepJDOT) define it by wasserstein distance based on the optimal transport problem (Damodaran et al., 2018). There are many variants, e.g. Convolutional deep Domain Adaptation model for Time Series data (CoDATS) is signal processing layers and multi sources version of DANNs (Wilson et al., 2020) and CoRALs variants are using different distant metrics for correlation alignment including a euclidean distance (Zhang et al., 2018a), geodesic distance (Zhang et al., 2018b), and log euclidean distance (Wang et al., 2017). For DANNs, optimizing $L_{domain}$ in a adversarial way with the domain discriminator $D$ with arbitrary neural networks' parameters $\theta_d$, we omitted description of $\arg\max_{\theta_d} L_{domain}$. Please note that $\arg\min_{\theta_c} L_{domain}$ is only worked for e.g. CoRALs and DANs.

We extend equation 1 to use intermediate data, as a decomposition of distribution differences between source and intermediate data and between intermediate and terminal target data, and same optimization can be used in this formula as well.

$$L_{propose} = L_{domain}(\mathcal{D}_S, \mathcal{D}_T) + L_{domain}(\mathcal{D}_T, \mathcal{D}_{T'}) \tag{6}$$

$$\mathrm{argmin}_{\theta_f, \theta_c}L_{task}, \mathrm{argmin}_{\theta_f, (\theta_c)}L_{propose} \tag{7}$$

The equation 6 says if we have intermediate data we can minimize $L_{domain}$ substituted by $L_{domain}(\mathcal{D}_S, \mathcal{D}_T)$ and $L_{domain}(\mathcal{D}_T, \mathcal{D}_{T'})$. Concurrent training of $L_{propose}, L_{task}$ is tantamount to **(1) aquisition of domain invariance between source and intermediate domain, discriminability for intermediate domain**, **(2) aqution of domain invariance between intermediate domain and target, discriminability for target**, we can do each quite easier compared with normal domain invariant learning between source and target with same goal due to data domain's semantic reason, so as a whole our strategy will facilitate learning target discriminability. Based on assumption of two dimensional data domains divergence between source and target is larger than divergence between source and intermediate, or divergence between intermediate and target, we can say $L_{domain} \geq \frac{L_{propose}}{2}$. This term is the optimisation target of domain invariant representation learning, but it can be inferred that the larger it is, the more likely it is to be addicted to stale solutions (e.g. local minima) when paralleled with loss minimisation for task classification. Pseudo codes for UDA with $L_{propose}$ are in Algorithm 1 and Algorithm 3 (Appendix A), though of course this can be used in other domain invariant representation learning techniques in the almost same way. We implemented two stages domain invariant learners with CoRALs as $L_{propose} = L_{domain}(\mathcal{D}_S, \mathcal{D}_T) + L_{domain}(\mathcal{D}_S, \mathcal{D}_{T'})$, since we can say same thing as $L_{propose} = L_{domain}(\mathcal{D}_S, \mathcal{D}_T) + L_{domain}(\mathcal{D}_T, \mathcal{D}_{T'})$ and to avoid noisy correlation alignment between intermediate and target data in the early stages of epochs. We denote $d_i$ as domain labels, $CE$ as the cross entropy loss, $BCE$ as the binary cross entropy loss, $MSE$ as the mean squared error and $Cov$ as the covariance matrix. The schematic diagram for DANNs version two stages domain invariant learners is in Figure 1 (CoRALs version in Appendix A Figure 7).

Previous paper (Oshima et al., 2024) is intuitively step-by-step version of this paper (center of Figure 1), this executes DANNs learning between source and intermediate data then second time DANNs between intermediate data (pseudo labeling by that learned task classifier) and target. We call this as Step-by-step. Very limited evaluation was conducted and achieved higher classification performance compared to normal DANNs and without adaptation model using occupancy detection data(Dataset

---

**Algorithm 1** 2stages-DANNs

---

**Require:** source,intermediate domain,target $\mathcal{D}_S, \mathcal{D}_T, \mathcal{D}_{T'}$
**Ensure:** neural network parameters $\{\theta_f, \theta_c, \theta_d, \theta_{d'}\}$

1: $\theta_f, \theta_c, \theta_d, \theta_{d'} \leftarrow \text{init}()$
2: **while** epoch_training() **do**
3:     **while** batch_training() **do**
4:         $\hat{\mathbb{E}}(L_{domain}(\mathcal{D}_S, \mathcal{D}_T)) \leftarrow$
        $\frac{1}{batch} \sum_{i=1}^{batch} \text{BCE}(D(F(x_i^S)), d_i^S) + \frac{1}{batch} \sum_{i=1}^{batch} \text{BCE}(D(F(x_i^T)), d_i^T)$
5:         $\hat{\mathbb{E}}(L_{domain}(\mathcal{D}_T, \mathcal{D}_{T'})) \leftarrow$
        $\frac{1}{batch} \sum_{i=1}^{batch} \text{BCE}(D(F(x_i^T)), d_i^T) + \frac{1}{batch} \sum_{i=1}^{batch} \text{BCE}(D(F(x_i^{T'})), d_i^{T'})$
6:         $\hat{\mathbb{E}}(L_{task}) \leftarrow \frac{1}{batch} \sum_{i=1}^{batch} \text{CE}(C(F(x_i^S)), y_i^S)$
7:         $\theta_c \leftarrow \theta_c - \frac{\partial \hat{\mathbb{E}}(L_{task})}{\partial \theta_c}$
8:         $\theta_f \leftarrow \theta_f - \frac{\partial(\hat{\mathbb{E}}(L_{task}) - \hat{\mathbb{E}}(L_{domain}(\mathcal{D}_S, \mathcal{D}_T)) - \hat{\mathbb{E}}(L_{domain}(T, T')))}{\partial \theta_f}$
9:         $\theta_d \leftarrow \theta_d - \frac{\partial \hat{\mathbb{E}}(L_{domain}(\mathcal{D}_S, \mathcal{D}_T))}{\partial \theta_d}$
10:      $\theta_{d'} \leftarrow \theta_{d'} - \frac{\partial \hat{\mathbb{E}}(L_{domain}(\mathcal{D}_T, \mathcal{D}_{T'}))}{\partial \theta_{d'}}$
11:     **end while**
12: **end while**

---

**Algorithm 2** two stages domain invariant learners free parameter indicator

---

**Require:** $\mathcal{D}_S, \mathcal{D}_T, \mathcal{D}_{T'}$, a neural network $\phi$

1: Split $\mathcal{D}_S$ into $\mathcal{D}_{S_{train}}$ and $\mathcal{D}_{S_{val}}$
2: Execute domain invariant learning with $\phi, \mathcal{D}_{S_{train}}, \mathcal{D}_T, \mathcal{D}_{T'}$, validate by $\mathcal{D}_{S_{val}}$ do early stopping
3: Pseudo labeling for $\mathcal{D}_{T'}$ get $\{(x_i^{T'}, \hat{y}_i^{T'})\}_{i=1}^{N_{T'}}$ using $\phi$
4: With pseudo-supervised $\{(x_i^{T'}, \hat{y}_i^{T'})\}_{i=1}^{N_{T'}}$ as source and $\mathcal{D}_T$ and unsupervised $\{x_i^S\}_{i=1}^{N_{S_{train}}}$, do domain invariant learning build $\phi_r$
5: calculate loss between predictions for $\mathcal{D}_{S_{val}}$ by $\phi_r$ and its ground truth labels

---

D, explain later). There are five main differences between the method of (Oshima et al., 2024) and this paper, (1) (Oshima et al., 2024) generates noise due to erroneous answers in the pseudo-labelling step ($\{(x_i^T, \hat{y}_i^T)\}_{i=1}^{N_T}$ in Figure 1), which may hinder learning convergence, whereas our end-to-end method does not, (2) The two domain invariant representation learnings have partially independent structure, and there is no guarantee that the first learning will necessarily be good for the second learning, but our method can perform the two learnings in end-to-end, which may make it easier to keep the overall balance, (3) They introduced confidence threshold technique which has large impact on learning convergence, but these engineering tricks are not easy to tune in unsupervised settings in practical use cases, (4) Comparative experiments on four sets of data show that our method has a performance advantage on most of the settings (5/8 settings) and (5) The learning algorithm encompasses the entire domain invariant representation learning techniques including CoRALs, DANNs and DANs.

## 3.2 FREE PARAMETERS TUNING

Free parameters(e.g. learning rate for gradient descent optimizer, layer-wise configurations) selection has been regarded as a crucial role because deep learning methods generally are susceptible to, additionally UDA require us to do this in a agnostic way for target ground truth labels at all. We propose free parameter tuning method specialized in this two stages domain invariant learners by extending RV. RV applies pseudo-labeling to the target data and uses the pseudo-labeled data and the source unsupervised data in an inverse relationship (Zhong et al., 2010), authors proved this can measure conditional distribution difference between target data and learned model. We apply the RV

idea to two stages domain invariant representation learners, replacing the internal learning steps from regular supervised machine learning with two stages domain invariant representation learners and calculating classification loss between predictions for source data by reverse model and its ground truth labels. We can find better configurations by $\arg\min_\theta |\phi_r(x^S) - y^S|$. Algorithm is denoted in Algorithm 2, and the Theorem 3.1 states that this method can measure the gap in labeling rules between the trained model and the final target data, the proof of the theorem is given in Appendix B.

**Theorem 3.1.** When executing Algorithm 2, conditional distribution gap between $\phi(\cdot)$ and $\mathcal{D}_{T'}$'s ground truth is calculated by ($C_1, C_2$ as constant values)

$$|\phi_r(x^S) - y^S| \propto |C_1\{P(y|x, \phi) - P_{T'}(y|x)\} + C_2\{P_T(y|x) - P_{T'}(y|x)\}| \tag{8}$$

## 4 EXPERIMENTAL VALIDATION

### 4.1 SETUP AND DATASETS

The evaluation follows the hold-out method. The evaluation score is the percentage of correct labels predicted by the task classifier of the two stages domain invariant learners for the target data i.e. $\text{accuracy}(x^{T'}) = \frac{1}{n}\sum_{x^{T'}} \mathbb{I}[C(F(x^{T'})) = y^{T'}]$ ($n$ as the sample size of target data for testing). The target data is divided into training data and test data in $50\%$, the training data is input to the training as unsupervised data, and the test data is not input to the training but used only when calculating the evaluation scores. In order to take into account the variations in the evaluation scores caused by the initial values of each layer of deep learning, 10 evaluations are carried out for each evaluation pattern and the average value is used as the final evaluation score. In addition, hyperparameters optimisation with 3.2 is performed on the learning rate using the training data. The all codes we used in this paper are available on GitHub [1].

We validate our method with 4 datasets and 38 tasks including simulated toy data, image digits recognition, HAR, occupancy detection. We adopt six benchmark models (conventional Train on Target model, Ste-by-step with DANNs or CoRALs, Normal with DANNs or CoRALs, conventional Without Adapt model) as a comparison test to ours in Table 14 (Appendix I). If our hypothesis is true, our method should be close to its Upper bound and better than previous studies and Lower bound models therefore ideal state "Upper bound > Ours > Max(Step-by-step, Normal, Lower bound)" should be expected.

**A. sklearn.datasets.make_moons**

Source data is two interleaving half circles with binary class. Intermediate data is rotated $a$ degrees (semi-clockwise from the centre), target data is rotated $2a$ degrees ($a \in \{15, 20, 25, 30, 35\}$). These rotations correspond to toy versions of two dimensional co-variate shifts.

**B. MNIST, MNIST-M, SVHN(Lecun et al., 1998; Ganin et al., 2017; Netzer et al., 2011)**

Source data is modified national institute of standards and technology database (MNIST), intermediate data is MNIST-M which is the randomly colored version of MNIST, target data is the street view house numbers (SVHN). Case with source and target reversed was demonstrated to be coped with by Normal (Ganin et al., 2017). Two dimensional co-variate shifts should be (Monotone, Not on the Street)→(Color, Not on the Street)→(Color, On the Street).

**C. HHAR(Stisen et al., 2015)**

Heterogeneity human activity recognition dataset (HHAR). Signal processing problem to determine human behaviour ($\{bike, sit, stand, walk, stairsup, stairsdown\}$) by accelerometer data (sliding window with size 128, sampling rate is 100-150Hz). Subjects were taking actions in the pre-programmed way, so there are no chances of distribution shift by different or strange movements. The two dimensional co-variate shifts should be (UserA, SensorA)→(UserB, SeonsorA)→(UserB, SensorB), same actions but different user and sensor differ in co-variate. We extract 16 patterns randomly from 9 users and 4 models.

---

[1] please check v1.0.0(release soon) compatible to this paper [Put URL later], also we elaborated experimental configurations in the Appendix D

**D. ECO data set**(Beckel et al., 2014)

Electricity consumption & occupancy (ECO) data set. Signal processing problem same as HHAR with different time window and activity classes(only $\{occupied, unoccupied\}$). The two dimensional co-variate shifts should be (HouseA, Winter)→(HouseB, Winter)→(HouseB, Summer). Subjects were not pre-programmed in any means so should include distribution shift in multiple ways, though basically the sharing of the rule of being at home when energy consumption is high and absent when it is not is indicated by (Oshima et al., 2024). 16 Patterns as total.

## 4.2 QUANTITATIVE RESULTS

We confirmed the Ours' obvious performance advantage compared to Step-by-step and Normal in Dataset A with DANNs when larger co-variate shift exists i.e. target data with 60 and 70 degrees rotated (highlighted in red in left of Figure 2). The difference was 0.174 compared to Normal when 60 degrees rotated target, 0.074 compared to Step-by-step and 0.091 compared to Normal when 70 degrees rotated target, and variance was much smaller. In Dataset A with CoRALs, increasing the angle of rotation significantly reduces the evaluation scores of Normal, but Step-by-step and Ours were somewhat able to cope with this (highlighted in red in right).

The Figure 3 shows our methods' superiority to previous studies in Dataset A-D(with DANNs), D(with CoRALs), namely the ideal state "Upper bound > Ours > Max(Step-by-step, Normal, Lower bound)" was observed in the UDA experiment. In cases Dataset A and C, a clear difference in accuracy was identified compared to Step-by-step or Normal, the difference in accuracy was 0.074 for Ours and Step-by-step in Dataset C, 0.061 for Ours and Normal, 0.053 for Ours and Normal in Dataset A. In cases other than the above, the degree of deviation from the ideal state is case-by-case. The superiority to Step-by-step i.e. "Ours > Step-by-step" is confirmed 5 out of 8 settings and this demonstrated effectiveness of proposing method i.e. end-to-end domain invariant learning with three sets of data. The superiority to Normal i.e. "Ours > Normal" is confirmed 8 out of 8 settings, highlighting the positive impact of two times domain invariant learnings itself.

Counting the cases where evaluation score is at least higher than the Lower bound, which is important in the UDA setting ("Ours > Lower bound"), 8/8 settings. This result means that when UDA is performed in business and real-world settings, a better discriminant model can be built than when it is not performed, highlighting its practical usefulness. Evaluation patterns and results for each Dataset, before aggregation, are provided in the Appendix C.

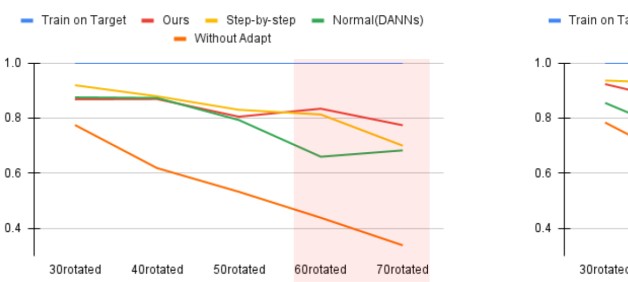 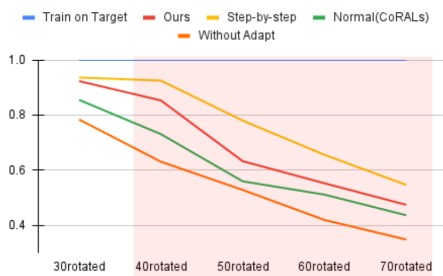

Figure 2: Comparison of the evaluation values between methods at each rotation angle in Dataset A (the left is with DANNs, right with CoRALs). Detailed results are in Appendix C Table 1 and 2 including the standard deviations.

## 4.3 QUALITATIVE RESULTS

To begin, we investigate how our method performs learning at each epoch in terms of domain invariance and task classifiability. We simultaneously visualise the learning loss of domain invariance per epoch (i.e. $-L_{domain} = CrossEntropy(\cdot)$), the learning loss of task classifiability and the synchronised evaluation of task classification on test target data. Figure 4 shows that in any case, domain invariance between the source and the intermediate domain and invariance between the in-

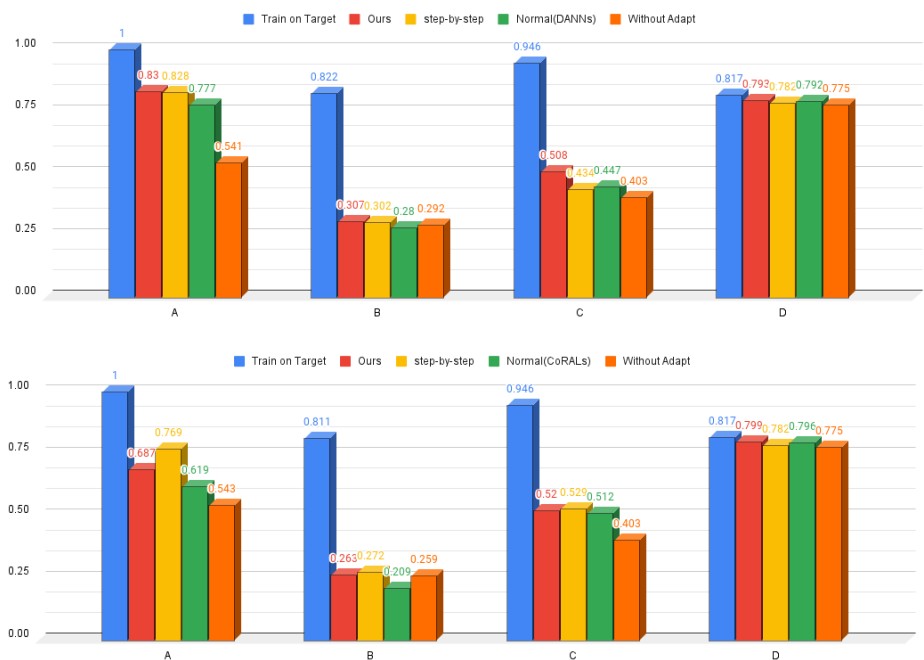

Figure 3: Quantitative result overview covering 8 methods and 4 datasets.

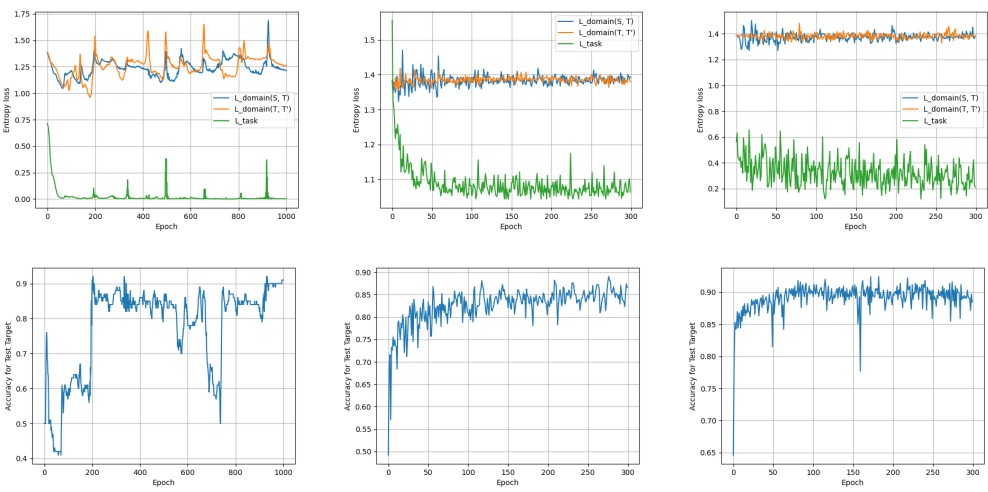

Figure 4: The $L_{task}, L_{domain}(\mathcal{D}_S, \mathcal{D}_T), L_{domain}(\mathcal{D}_T, \mathcal{D}_{T'})$ and evaluation per epoch. The $L_{domain}(\mathcal{D}_S, \mathcal{D}_T)$ as $L_{domain}(S, T)$ for short. Column is one trial for Dataset A (source→30rotated→60rotated), C ((d,s3mini)→(e,s3)), D ((3, s)→(1, w)) with DANNs.

termediate domain and the target are learnt in an adversarial manner and eventually a solution with high invariance is reached. Also we found that the task classifiability to the source is simultaneously optimised and eventually asymptotically approaches zero or small value. Correspondingly to the above three learnings, the evaluation scores are improving and we can recognise that our proposed optimisation algorithm is effective in the point of task classification performance for target data.

To get more insights into our method, we investigated neural networks' learned representation at both of feature level and classifier level. The Figure 5 includes feature extractor's learned representation, task classifier's sigmoid probability for grid space and representation at feature level is

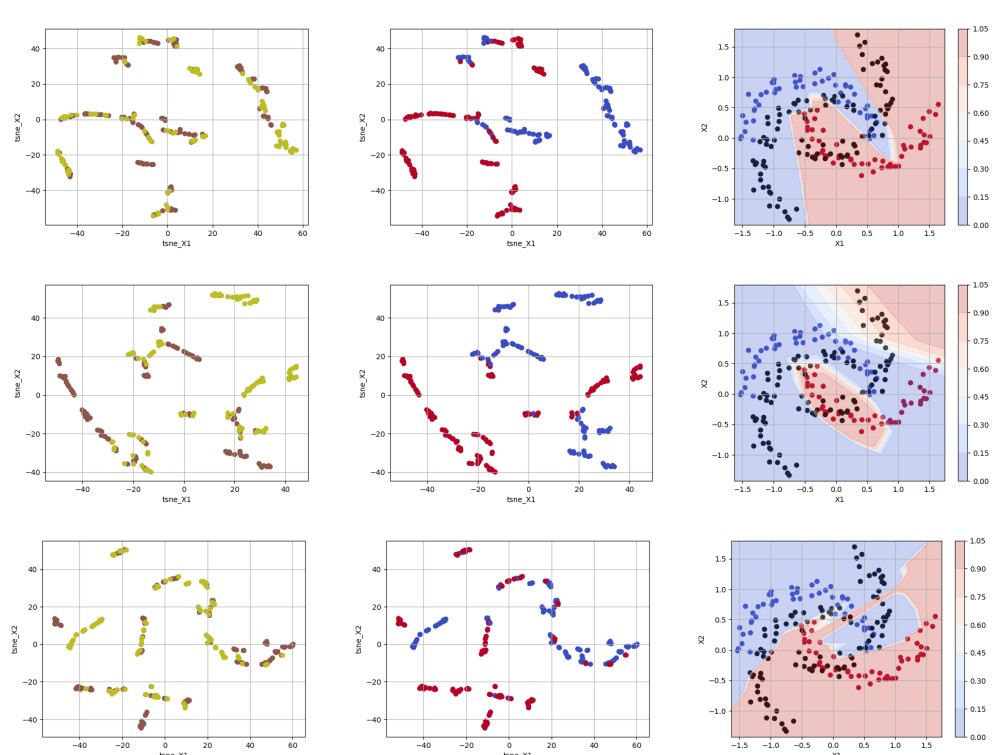

Figure 5: Learned representation at different levels in the Dataset A (source→30rotated→60rotated) experiment with DANNs. The first column corresponds to feature representation with domain labels color, second feature representation with task labels and third one is predictive probability for grid space. Rows express methods(Ours, Step-by-step, Normal in order). Representations were gone through t-distributed stochastic neighbor embedding (t-SNE)(van der Maaten & Hinton, 2008).

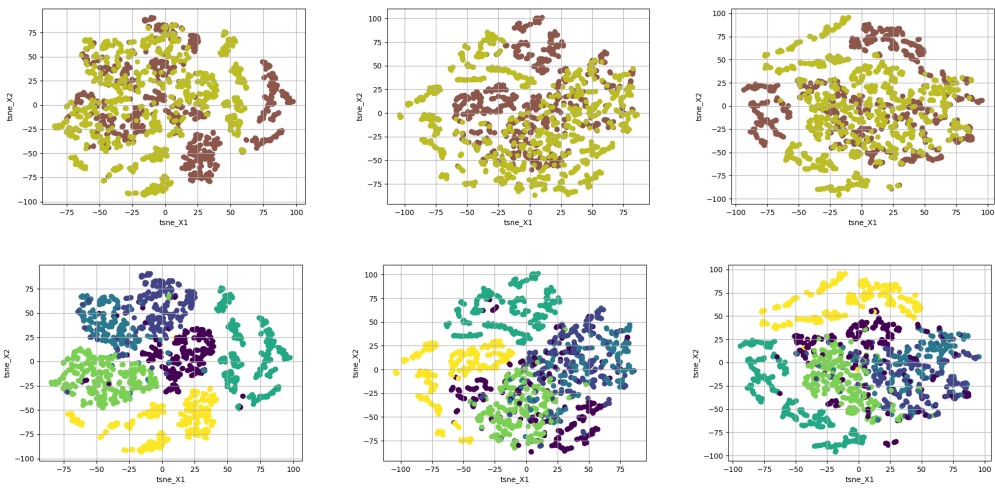

Figure 6: Learned representation at feature level in one trial from Dataset C ((d,s3mini)→(e,s3)) with DANNs. The first row corresponds to representation with domain labels, second feature representation with task labels. Columns represent methods(Ours, Step-by-step, Normal in order).

colored in two ways i.e. domain labels and task labels. We can qualitatively recognize ours could learn domain invariant feature between source and target and task discriminability for target data while keeping balance between both. Even though we have not accessed any labels for the target data in black in the figure, we can see a probability boundary that discriminates the data approximately perfectly. On the other hand Normal and Step-by-step could not. Normal could not get domain invariance, task variance, and appropriate boundary. Step-by-step could not learn domain invariance and appropriate boundary.

The Figure 6 shows Ours' success of domain invariant and task variant features acquirement. Other methods are struggling to get task variant features e.g. the space around the yellow-green scatter points is very dense, including the other three classes. It is difficult for them to classify these human's behaviour classes (this four classes are $\{bike, walk, stairsup, stairsdown\}$ far from $\{stand, sit\}$, Ours could overcome this apparently). The Step-by-step could not learn domain invariant features since we can very easily identify yellow and brown scatter points at a glance.

## 5 CONCLUSION AND FUTURE RESEARCH DIRECTION

We proposed novel UDA strategy whose domain invariance and task variant nature could overcome UDA problem with two dimensional co-variate shifts. Also our proposing free parameters tuning method is useful since it can validate UDA model automatically without access to target ground truth labels.

In terms of research directions for the method, further improvements in accuracy can be expected when the method is used in combination with other UDA methods e.g. (Yang et al., 2024; French et al., 2018; Sun et al., 2022; Yang et al., 2021; Singha et al., 2023). Our method is attractive because it is a broad abstraction that encompasses layers of deep learning and domain invariant representation learning methods internally, and can be used in combination with many other methods (not limited to specific data type e.g. table data, image, and signal). The hyper-parameter optimisation method of (Yang et al., 2024) uses not the conditional distribution gap that can be measured by this method, but with the hopkins statistics (Banerjee & Dave, 2004) and mutual information, transferability at classifier level and transferability and discriminability at feature level can be measured in a combined manner. Also (French et al., 2018) is based on that consistency regularisation ensures that the neural networks' outputs are close each other for stochastic perturbations. It is well-matched with tons of image augmentation methods of image processing (Shorten & Khoshgoftaar, 2019) and is likely to improve experiments with Dataset B in particular. Safe Self-Refinement for Transformer-based domain adaptation uses vision transformer backbone and consistency regularization as well (Sun et al., 2022), might improve the performance (vision transformer based UDA was also analyzed in another paper (Yang et al., 2021) and improved the performance). Also large vision-language models' prompting was used for UDA. They format domain invariant and task variant information as the prompting and showed impressive performance improvements in image recognition datasets(Singha et al., 2023).

Another research question is how to obtain an intermediate domain. For time series data, databases normally hold huge amounts of unsupervised data(He et al., 2015; Wang et al., 2019a), so two dimensional co-variate shifts assumption is quite natural. For other data types such as image data, your database can't always hold a suitable intermediate. One way is to use the Web with huge unlabeled data. Alto this should be a future research direction e.g. generative models possibly will create suitable one based on prompting or papers (Lin et al., 2021; Zhang et al., 2019) methods might help create intermediate synthetically.

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

## A    CoRALs version two stages domain invariant learners

The pseudo code for Ours with CoRALs is in Algorithm 3 and the schematic diagram is in Figure 7.

---

**Algorithm 3** 2stages-CoRALs

---

**Require:** source,intermediate domain,target $\mathcal{D}_S, \mathcal{D}_T, \mathcal{D}_{T'}$
**Ensure:** neural network parameters $\{\theta_f, \theta_c\}$
1: $\theta_f, \theta_c \leftarrow \text{init}()$
2: **while** epoch_training() **do**
3:    **while** batch_training() **do**
4:       $\hat{\mathbb{E}}(L_{domain}(\mathcal{D}_S, \mathcal{D}_T)) \leftarrow$
          $\frac{1}{batch} \sum_{i=1}^{batch} MSE(Cov(C(F(x_i^S))), Cov(C(F(x_i^T))))$
5:       $\hat{\mathbb{E}}(L_{domain}(\mathcal{D}_S, \mathcal{D}_{T'})) \leftarrow$
          $\frac{1}{batch} \sum_{i=1}^{batch} MSE(Cov(C(F(x_i^S))), Cov(C(F(x_i^{T'}))))$
6:       $\hat{\mathbb{E}}(L_{task}) \leftarrow \frac{1}{batch} \sum_{i=1}^{batch} CE(C(F(x_i^S)), y_i^S)$
7:       $\theta_c \leftarrow \theta_c - \frac{\partial(\hat{\mathbb{E}}(L_{task}) + \hat{\mathbb{E}}(L_{domain}(\mathcal{D}_S, \mathcal{D}_T)) + \hat{\mathbb{E}}(L_{domain}(\mathcal{D}_S, \mathcal{D}_{T'})))}{\partial \theta_c}$
8:       $\theta_f \leftarrow \theta_f - \frac{\partial(\hat{\mathbb{E}}(L_{task}) + \hat{\mathbb{E}}(L_{domain}(\mathcal{D}_S, \mathcal{D}_T)) + \hat{\mathbb{E}}(L_{domain}(\mathcal{D}_S, \mathcal{D}_{T'})))}{\partial \theta_f}$
9:    **end while**
10: **end while**

---

## B    Proof of Reverse Validation based free parameters tuning

Line 2 in Algorithm 2 is saying learned $\phi$ should be approximation of mixing of $P_S(y|x), P_T(y|x), P_{T'}(y|x)$ and line 4 can be seen in the same way. We omitted the descriptions of approximation errors.

$$P(y|x, \phi) = (1 - \gamma - \delta)P_S(y|x) + \gamma P_T(y|x) + \delta P_{T'}(y|x)$$

$$P(y|x, \phi_r) = (1 - \alpha - \beta)P(y|x, \phi) + \alpha P_S(y|x) + \beta P_T(y|x)$$

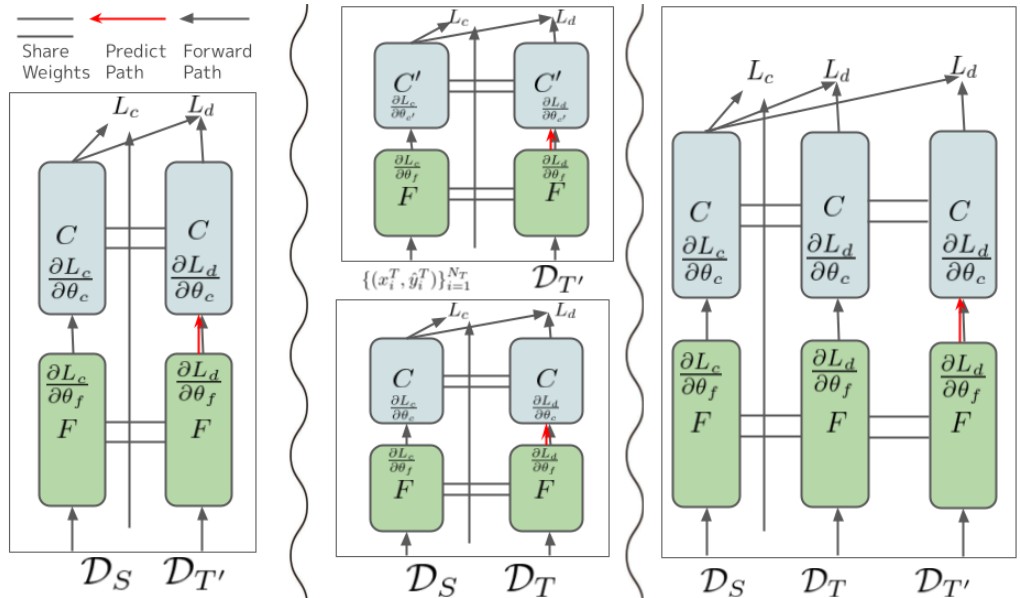

Figure 7: Forward path and backward process of Normal(CoRALs), Step-by-step, Ours. The $L_{task}, L_{domain}$ as $L_c, L_d$ for short.

Where $\alpha, \beta, \gamma, \delta$ are the nuisance parameters related to the ratio between the sizes of distributions.

$$
\begin{aligned}
|\phi_r(x^S) - y^S| &= |P(y|x, \phi_r) - P_S(y|x)| \\
&= |(1 - \alpha - \beta)P(y|x, \phi) + \alpha P_S(y|x) + \beta P_T(y|x) - P_S(y|x)| \\
&= |(1 - \alpha - \beta)P(y|x, \phi) + \beta P_T(y|x) - \frac{1-\alpha}{1-\gamma-\delta}\{P(y|x, \phi) - \\
&\quad \gamma P_T(y|x) - \delta P_{T'}(y|x)\}| \\
|\phi_r(x^S) - y^S|(1 - \gamma - \delta) &= |\{(1 - \alpha - \beta)P(y|x, \phi) + \beta P_T(y|x) - \frac{1-\alpha}{1-\gamma-\delta}\{P(y|x, \phi) - \\
&\quad \gamma P_T(y|x) - \delta P_{T'}(y|x)\}\}(1 - \gamma - \delta)| \\
&= |C_1 P(y|x, \phi) + C_2 P_T(y|x) + C_3 P_{T'}(y|x)| \\
&= |C_1 P(y|x, \phi) + \{C_2 + C_3\}P_{T'}(y|x) + C_2\{P_T(y|x) - P_{T'}(y|x)\}| \\
|\phi_r(x^S) - y^S| &= |C_1\{P(y|x, \phi) - P_{T'}(y|x)\} + C_2\{P_T(y|x) - P_{T'}(y|x)\}|C_4
\end{aligned}
$$

We denoted fixed values as $C_1, C_2, C_3, C_4$ respectively.$C_1 = (1 - \alpha - \beta)(1 - \gamma - \delta) - (1 - \alpha)$, $C_2 = \beta(1 - \gamma - \delta) + (1 - \alpha)\gamma$, $C_3 = (1 - \alpha)\delta$, $C_4 = \frac{1}{1-\gamma-\delta}$

## C  QUANTITATIVE RESULTS IN DETAIL

Detailed results including the pattern of each data, each method and each domain adaptation task(as PAT for short in these tables) before calculating the average for each dataset and method are in Table 2-13.

## D  IMPLEMENTATION CONFIGURATION IN DETAIL

Data load and pre-processing steps are same as UDA previous studies (Ganin et al., 2017; Wilson et al., 2020; Oshima et al., 2024) (experiment with Dataset A and B from (Ganin et al., 2017), Dataset C from (Wilson et al., 2020), Dataset D from (Oshima et al., 2024)). For standardisation pre-processing, the statistics of the target test data are not accessed, only the statistics of the training data are accessed and executed. Internal layers are same between methods, Normal and Step-by-step and

Table 1: Dataset A with DANNs. The best method in each PAT except for Train on Target was specified in bold. The value inside of parentheses is the standard deviation of 10 times evaluations, we omitted the depictions for Dataset B-D.

| PAT | Train on Target | Ours | Step-by-step | Normal(DANNs) | Without Adapt |
|---|---|---|---|---|---|
| 15rotated→30rotated | 1(0) | 0.868(.07) | **0.919**(.06) | 0.875(.07) | 0.775(.02) |
| 20rotated→40rotated | 1(0) | 0.869(.08) | **0.879**(.09) | 0.873(.06) | 0.619(.03) |
| 25rotated→50rotated | 1(0) | 0.805(.06) | **0.830**(.1) | 0.793(.1) | 0.533(.02) |
| 30rotated→60rotated | 1(0) | **0.834**(.04) | 0.813(.07) | 0.660(.2) | 0.439(.05) |
| 35rotated→70rotated | 1(0) | **0.774**(.1) | 0.700(.1) | 0.683(.2) | 0.339(.01) |
| Average | 1(0) | **0.830**(.07) | 0.828(.09) | 0.777(.1) | 0.541(.03) |

Table 2: Dataset A with CoRALs.

| PAT | Train on Target | Ours | Step-by-step | Normal(CoRALs) | Without Adapt |
|---|---|---|---|---|---|
| 15rotated→30rotated | 1(0) | 0.923(.06) | **0.936**(.07) | 0.855(.1) | 0.784(.04) |
| 20rotated→40rotated | 1(0) | 0.853(.09) | **0.925**(.07) | 0.731(.09) | 0.631(.03) |
| 25rotated→50rotated | 1(0) | 0.633(.1) | **0.780**(.1) | 0.560(.07) | 0.529(.03) |
| 30rotated→60rotated | 1(0) | 0.553(.2) | **0.656**(.2) | 0.512(.06) | 0.420(.02) |
| 35rotated→70rotated | 1(0) | 0.475(.1) | **0.547**(.1) | 0.437(.2) | 0.349(.01) |
| Average | 1(0) | 0.687(.1) | **0.769**(.1) | 0.619(.08) | 0.543(.03) |

Table 3: Dataset A with JDOT.

| PAT | Train on Target | Ours | Step-by-step | Normal(JDOT) | Without Adapt |
|---|---|---|---|---|---|
| 15rotated→30rotated | 1(0) | **0.934**(.02) | 0.895(.07) | 0.8882(.04) | 0.782(.04) |
| 20rotated→40rotated | 1(0) | **0.839**(.08) | 0.764(.05) | 0.787(.04) | 0.617(.03) |
| 25rotated→50rotated | 1(0) | **0.720**(.1) | 0.677(.04) | 0.707(.09) | 0.530(.04) |
| 30rotated→60rotated | 1(0) | **0.620**(.1) | 0.522(.03) | 0.554(.1) | 0.427(.02) |
| 35rotated→70rotated | 1(0) | **0.492**(.1) | 0.440(.08) | 0.434(.05) | 0.329(.01) |
| Average | 1(0) | **0.721**(.09) | 0.660(.05) | 0.673(.06) | 0.537(.03) |

Table 4: Dataset B with DANNs.

| Trial index | Train on Target | Ours | Step-by-step | Normal(DANNs) | Without Adapt |
|---|---|---|---|---|---|
| 0 | 0.8142286539077759 | 0.2569529712200165 | **0.31246158480644226** | 0.2764674127101898 | 0.2764674127101898 |
| 1 | 0.8248693943023682 | **0.29966962337493896** | 0.29859402775764465 | 0.29452213644981384 | 0.288836807012558 |
| 2 | 0.8270590305328369 | **0.365281194448471** | 0.29548248648643494 | 0.2744314670562744 | 0.3106561303138733 |
| Average | 0.8220523596 | **0.307301263014475** | 0.302179366350173 | 0.28019360701243 | 0.29198678334554 |

Table 5: Dataset B with CoRALs.

| Trial index | Train on Target | Ours | Step-by-step | Normal(CoRALs) | Without Adapt |
|---|---|---|---|---|---|
| 0 | 0.824062705039978 | 0.28626304864883423 | **0.2970958948135376** | 0.1885755956172943 | 0.2821911573410034 |
| 1 | 0.8216425776481628 | **0.27185770869255066** | 0.267324835062027 | 0.2312154322862625 | 0.23025506734848022 |
| 2 | 0.7872233986854553 | 0.231292262673378 | 0.2524969279766083 | 0.20732176303863525 | **0.2641364336013794** |
| Average | 0.810976227124532 | 0.263137673338254 | **0.272305885950724** | 0.20903759698073 | 0.258860886096954 |

Table 6: Dataset B with JDOT.

| Trial index | Train on Target | Ours | Step-by-step | Normal(JDOT) | Without Adapt |
|---|---|---|---|---|---|
| 0 | 0.766364455223083 | **0.275046110153198** | 0.252228021621704 | 0.252727419137954 | 0.274469882249832 |
| 1 | 0.829706487655639 | 0.281576514244079 | 0.303741544485092 | 0.273547947406768 | **0.28322833776474** |
| 2 | 0.763560235500335 | **0.288875222206115** | 0.23782268166542 | 0.213429629802703 | 0.258681625127792 |
| Average | 0.786543726126352 | **0.281832615534464** | 0.264597415924072 | 0.246568332115808 | 0.272126615047455 |

Table 7: Dataset C with DANNs.

| PAT | Train on Target | Ours | Step-by-step | Normal(DANNs) | Without Adapt |
|---|---|---|---|---|---|
| (d,nexus4)→(f,samsungold) | 0.979234308 | 0.333990708 | **0.3791183114** | 0.2445475519 | 0.3040603146 |
| (f,s3mini)→(g,s3) | 0.9537375212 | **0.6039866924** | 0.3563122801 | 0.4890365332 | 0.4035714209 |
| (d,s3mini)→(e,s3) | 0.9632268667 | **0.8621621609** | 0.6398853391 | 0.6870597839 | 0.8511875629 |
| (b,s3)→(f,s3mini) | 0.9091445386 | **0.7703539729** | 0.6380531043 | 0.6666666627 | 0.7166666567 |
| (a,nexus4)→(d,s3) | 0.9471365869 | **0.5560352564** | 0.4770925194 | 0.5336563945 | 0.3418502301 |
| (d,s3)→(e,samsungold) | 0.9668659866 | 0.3383971155 | **0.3572966397** | 0.3135167331 | 0.2260765433 |
| (e,s3mini)→(i,nexus4) | 0.9696132898 | **0.633001852** | 0.5512707353 | 0.5638305902 | 0.5797790289 |
| (e,samsungold)→(f,s3) | 0.9335558951 | **0.4998330414** | 0.4212019861 | 0.4297161847 | 0.3644407243 |
| (f,samsungold)→(h,s3) | 0.9565408528 | 0.230904308 | 0.2175592646 | 0.2302897334 | **0.2808604091** |
| (f,s3mini)→(g,nexus4) | 0.9656078279 | **0.5516470492** | 0.3412548937 | 0.4018039137 | 0.3747843146 |
| (b,samsungold)→(h,s3mini) | 0.8693650961 | **0.2890476286** | 0.2026984192 | 0.2434920691 | 0.2863492191 |
| (c,s3)→(i,nexus4) | 0.9692449689 | 0.5139595062 | **0.5257458717** | 0.5067403525 | 0.2813628078 |
| (a,nexus4)→(e,s3mini) | 0.8998363554 | **0.5181669414** | 0.512438634 | 0.5076923162 | 0.3450081885 |
| (h,s3)→(i,nexus4) | 0.9661510408 | 0.5531123698 | 0.5488398015 | **0.558121568** | 0.476427266 |
| (b,nexus4)→(e,s3mini) | 0.9135843039 | **0.588052386** | 0.5351882339 | 0.508019653 | 0.4073649794 |
| (a,s3)→(b,samsungold) | 0.9768714905 | **0.2908379823** | 0.2344133988 | 0.2617877066 | 0.2089385405 |
| Average | 0.9462323081 | **0.5083430607** | 0.4336480896 | 0.4466236092 | 0.4030455129 |

Table 8: Dataset C with CoRALs.

| PAT | Train on Target | Ours | Step-by-step | Normal(CoRALs) | Without Adapt |
|---|---|---|---|---|---|
| (d,nexus4)→(f,samsungold) | 0.979234308 | 0.2712296873 | **0.3948955804** | 0.2451276004 | 0.3040603146 |
| (f,s3mini)→(g,s3) | 0.9537375212 | 0.5656976521 | **0.5835548103** | 0.5803986609 | 0.4035714209 |
| (d,s3mini)→(e,s3) | 0.9632268667 | 0.8188370228 | 0.7868959963 | 0.8158067286 | **0.8511875629** |
| (b,s3)→(f,s3mini) | 0.9091445386 | 0.7690265477 | 0.7792035401 | **0.7799409986** | 0.7166666567 |
| (a,nexus4)→(d,s3) | 0.9471365869 | **0.5544493496** | 0.5429075032 | 0.529603532 | 0.3418502301 |
| (d,s3)→(e,samsungold) | 0.9668659866 | 0.3571770191 | 0.2886363477 | **0.3734449655** | 0.2260765433 |
| (e,s3mini)→(i,nexus4) | 0.9696132898 | 0.6311602473 | **0.7156169653** | 0.5859300375 | 0.5797790289 |
| (e,samsungold)→(f,s3) | 0.9335558951 | 0.5492487252 | **0.5727879584** | 0.52420699 | 0.3644407243 |
| (f,samsungold)→(h,s3) | 0.9565408528 | 0.2807726175 | 0.2603160739 | 0.274363485 | **0.2808604091** |
| (f,s3mini)→(g,nexus4) | 0.9656078279 | 0.5189019471 | 0.4879607767 | **0.567921567** | 0.3747843146 |
| (b,samsungold)→(h,s3mini) | 0.8693650961 | **0.2895238206** | 0.2641269937 | 0.2858730286 | 0.2863492191 |
| (c,s3)→(i,nexus4) | 0.9692449689 | 0.6111602366 | **0.6216574728** | 0.5978268981 | 0.2813628078 |
| (a,nexus4)→(e,s3mini) | 0.8998363554 | 0.5220949322 | 0.5574468166 | **0.639607209** | 0.3450081885 |
| (h,s3)→(i,nexus4) | 0.9661510408 | 0.5809944987 | **0.5965377748** | 0.543167603 | 0.476427266 |
| (b,nexus4)→(e,s3mini) | 0.9135843039 | 0.7222586095 | **0.7522095025** | 0.5345335573 | 0.4073649794 |
| (a,s3)→(b,samsungold) | 0.9768714905 | 0.2730726153 | 0.2655865803 | **0.3111731768** | 0.2089385405 |
| Average | 0.9462323081 | 0.5197253455 | **0.529199034** | 0.5120051367 | 0.4030455129 |

Table 9: Dataset C with JDOT.

| PAT | Train on Target | Ours | Step-by-step | Normal(JDOT) | Without Adapt |
|---|---|---|---|---|---|
| (d,nexus4)→(f,samsungold) | 0.979234308 | **0.355220401287078** | 0.317865419387817 | 0.321693730354309 | 0.3040603146 |
| (f,s3mini)→(g,s3) | 0.9537375212 | **0.503405305743217** | 0.438787361979484 | 0.503405302762985 | 0.4035714209 |
| (d,s3mini)→(e,s3) | 0.9632268667 | 0.851760858297348 | **0.866830480098724** | 0.864209669828415 | 0.8511875629 |
| (b,s3)→(f,s3mini) | 0.9091445386 | 0.74144542813301 | 0.73274335861206 | **0.754129791259765** | 0.7166666567 |
| (a,nexus4)→(d,s3) | 0.9471365869 | 0.389691638946533 | 0.348017627000808 | **0.392158597707748** | 0.3418502301 |
| (d,s3)→(e,samsungold) | 0.9668659866 | **0.339234438538551** | 0.336124384403228 | 0.33349280655384 | 0.2260765433 |
| (e,s3mini)→(i,nexus4) | 0.9696132898 | 0.58184163570404 | **0.605451226234436** | 0.588397806882858 | 0.5797790289 |
| (e,samsungold)→(f,s3) | 0.9335558951 | **0.534557574987411** | 0.497829702496528 | 0.527045065164566 | 0.3644407243 |
| (f,samsungold)→(h,s3) | 0.9565408528 | **0.310798954963684** | 0.146356455236673 | 0.268832318484783 | 0.2808604091 |
| (f,s3mini)→(g,nexus4) | 0.9656078279 | **0.446235281229019** | 0.332862740755081 | 0.438745093345642 | 0.3747843146 |
| (b,samsungold)→(h,s3mini) | 0.8693650961 | 0.287936517596244 | **0.298888900876045** | 0.278095249831676 | 0.2863492191 |
| (c,s3)→(i,nexus4) | 0.9692449689 | **0.376316770911216** | 0.30895028412342 | 0.308692456781864 | 0.2813628078 |
| (a,nexus4)→(e,s3mini) | 0.8998363554 | 0.32962357699871 | **0.376268419623374** | 0.335024553537368 | 0.3450081885 |
| (h,s3)→(i,nexus4) | 0.9661510408 | **0.539668530225753** | 0.493407014012336 | 0.518563541769981 | 0.476427266 |
| (b,nexus4)→(e,s3mini) | 0.9135843039 | 0.377741411328315 | **0.53698855638504** | 0.38936171233654 | 0.4073649794 |
| (a,s3)→(b,samsungold) | 0.9768714905 | **0.264245799183845** | 0.202234633266925 | 0.258994407951831 | 0.2089385405 |
| Average | 0.9462323081 | **0.451857757754623** | 0.427475410280749 | 0.442552631534636 | 0.4030455129 |

Table 10: Dataset D with DANNs.

| PAT | Train on Target | Ours | Step-by-step | Normal(DANNs) | Without Adapt |
|---|---|---|---|---|---|
| (1, w)→(2, s) | 0.8957642913 | 0.7409972489 | **0.7875346422** | 0.7360110939 | 0.6867036104 |
| (1, w)→(3, s) | 0.861866653 | 0.7052208841 | **0.7192771018** | 0.6852744341 | 0.6829986632 |
| (2, w)→(1, s) | 0.8012861729 | 0.693053323 | **0.7054927468** | 0.6919224679 | 0.6621971011 |
| (2, w)→(3, s) | 0.8601333261 | 0.7986613035 | 0.7954484522 | 0.8053547502 | **0.8078982592** |
| (3, w)→(1, s) | 0.807395494 | 0.6822294176 | **0.7321486413** | 0.6969305456 | 0.6933764279 |
| (3, w)→(2, s) | 0.8927256107 | 0.7096952975 | **0.7237303913** | 0.696583581 | 0.6832871735 |
| (4, w)→(5, s) | 0.8287172318 | 0.8516837239 | **0.853587091** | 0.8380673289 | 0.81859442 |
| (5, w)→(4, s) | 0.8698020101 | 0.876616919 | **0.8845771074** | 0.8573797703 | 0.8353233814 |
| (1, s)→(2, w) | 0.8868200541 | 0.8035789073 | 0.8214736462 | 0.8545262694 | **0.8671578526** |
| (1, s)→(3, w) | 0.780769217 | 0.7954063714 | 0.66077739 | **0.8070671439** | 0.7749116659 |
| (2, s)→(1, w) | 0.6769754767 | **0.8073871732** | 0.7716826499 | 0.8065663874 | 0.7937072694 |
| (2, s)→(3, w) | 0.7828671217 | 0.7583038926 | 0.7551236808 | 0.760777396 | **0.7763250887** |
| (3, s)→(1, w) | 0.724523145 | 0.8347469509 | 0.7835841656 | **0.8384405255** | 0.8228454411 |
| (3, s)→(2, w) | 0.8864016414 | 0.8627368033 | 0.8244210184 | 0.8679999769 | **0.8871578455** |
| (4, s)→(5, w) | 0.7400809884 | 0.8338086188 | 0.7645621732 | 0.8256619632 | **0.84969455** |
| (5, s)→(4, w) | 0.7781984448 | **0.9292267561** | **0.9292267561** | 0.9102228284 | 0.7644823313 |
| Average | 0.8171454299 | **0.7927095994** | 0.7820404784 | 0.7924241539 | 0.7754163176 |

Table 11: Dataset D with CoRALs.

| PAT | Train on Target | Ours | Step-by-step | Normal(CoRALs) | Without Adapt |
|---|---|---|---|---|---|
| (1, w)→(2, s) | 0.8957642913 | 0.7522622466 | **0.7524469137** | 0.7475531042 | 0.6867036104 |
| (1, w)→(3, s) | 0.861866653 | 0.7228915513 | 0.7121820569 | **0.7263721406** | 0.6829986632 |
| (2, w)→(1, s) | 0.8012861729 | **0.6936995268** | 0.6898223042 | 0.6877221525 | 0.6621971011 |
| (2, w)→(3, s) | 0.8601333261 | 0.8285140514 | 0.8327978611 | **0.8334672034** | 0.8078982592 |
| (3, w)→(1, s) | 0.807395494 | 0.7008077681 | **0.7260097086** | 0.6957996964 | 0.6933764279 |
| (3, w)→(2, s) | 0.8927256107 | 0.7213296473 | 0.7214219868 | **0.7216989934** | 0.6832871735 |
| (4, w)→(5, s) | 0.8287172318 | **0.853587091** | **0.853587091** | **0.853587091** | 0.81859442 |
| (5, w)→(4, s) | 0.8698020101 | 0.8825870633 | **0.8850746214** | 0.8812603652 | 0.8353233814 |
| (1, s)→(2, w) | 0.8868200541 | 0.831789434 | 0.8061052263 | 0.7663157582 | **0.8671578526** |
| (1, s)→(3, w) | 0.780769217 | **0.7992932916** | 0.7176678598 | 0.7975265026 | 0.7749116659 |
| (2, s)→(1, w) | 0.6769754767 | **0.7957592607** | 0.7577291667 | 0.7726402462 | 0.7937072694 |
| (2, s)→(3, w) | 0.7828671217 | 0.7575971723 | 0.6526501805 | 0.7731448889 | **0.7763250887** |
| (3, s)→(1, w) | 0.724523145 | 0.8332421601 | 0.8025992155 | **0.8411765158** | 0.8228454411 |
| (3, s)→(2, w) | 0.8864016414 | 0.8532631218 | 0.8418946981 | 0.8774736524 | **0.8871578455** |
| (4, s)→(5, w) | 0.7400809884 | 0.8350306153 | 0.8350306153 | 0.8350306153 | **0.84969455** |
| (5, s)→(4, w) | 0.7781984448 | 0.9263434052 | **0.9292267561** | 0.9259502232 | 0.7644823313 |
| Average | 0.8171454299 | **0.7992498379** | 0.7822653914 | 0.7960449468 | 0.7754163176 |

Table 12: Dataset D with JDOT.

| PAT | Train on Target | Ours | Step-by-step | Normal(JDOT) | Without Adapt |
|---|---|---|---|---|---|
| (1, w)→(2, s) | 0.8957642913 | 0.73711912035942 | **0.7653739690780** | 0.717543876171112 | 0.6867036104 |
| (1, w)→(3, s) | 0.861866653 | 0.70575635433197 | **0.7164658486843** | 0.696385538578033 | 0.6829986632 |
| (2, w)→(1, s) | 0.8012861729 | **0.7024232804723** | 0.691599369049072 | 0.686106634140014 | 0.6621971011 |
| (2, w)→(3, s) | 0.8601333261 | **0.8397590339188** | 0.829183393716812 | 0.810575628280639 | 0.8078982592 |
| (3, w)→(1, s) | 0.807395494 | 0.689499223232269 | **0.7218093872031** | 0.690468513965606 | 0.6933764279 |
| (3, w)→(2, s) | 0.8927256107 | 0.705632507801055 | **0.7588181078499** | 0.710249316692352 | 0.6832871735 |
| (4, w)→(5, s) | 0.8287172318 | 0.853294265270233 | **0.8535870909690** | 0.851098072528839 | 0.81859442 |
| (5, w)→(4, s) | 0.8698020101 | 0.866003310680389 | 0.791708122938871 | **0.8747927069664** | 0.8353233814 |
| (1, s)→(2, w) | 0.8868200541 | 0.82652627825737 | 0.815578907728195 | 0.853052592277526 | **0.8671578526** |
| (1, s)→(3, w) | 0.780769217 | **0.7865724503998** | 0.74770318865776 | 0.763250893354415 | 0.7749116659 |
| (2, s)→(1, w) | 0.6769754767 | **0.8117647409408** | 0.786183339357376 | 0.810807144641876 | 0.7937072694 |
| (2, s)→(3, w) | 0.7828671217 | 0.768551242351531 | **0.7830388784407** | 0.764664322137832 | 0.7763250887 |
| (3, s)→(1, w) | 0.724523145 | **0.8399453103542** | 0.777428218722343 | 0.808344769477844 | 0.8228454411 |
| (3, s)→(2, w) | 0.8864016414 | 0.867789441347122 | 0.855368375778198 | 0.863999962806701 | **0.887157845** |
| (4, s)→(5, w) | 0.7400809884 | 0.838900262117385 | 0.835030615329742 | 0.832179284095764 | **0.84969455** |
| (5, s)→(4, w) | 0.7781984448 | 0.832503297179937 | **0.91310617923** | 0.871428591012954 | 0.7644823313 |
| Average | 0.8171454299 | **0.7920025074388** | 0.790123937046155 | 0.787809240445494 | 0.7754163176 |

Ours have same layers and same shape of $F, C, D$ respectively though the number of components is not the same since e.g. Normal does not have two domain discriminators. The number of $F, C$ when inference is same. Internal layers are from previous studies, Dataset A with shallow neural networks is from (Ganin et al., 2017), Dataset B with CNN based back bone from (Ganin et al., 2017), Dataset C with one dimensional CNN based from Figure 3 in (Wilson et al., 2020) and Dataset D from Figure 4 and section 4.3. in (Oshima et al., 2024). In the settings during learning, a fixed learning rate is adopted for Dataset A and B. For the other Datasets, the learning rate is determined by optimising from 0.001-0.00001 using the Theorem 3.1 method. In Step-by-step and Normal, the (Ganin et al., 2017) method is used to perform the optimisation. Terminal Evaluation step for target data is exactly same for any method, the feature extractor and task classifier are applied to the 50% of target data not used for training in any sense and their predictions are compared with the ground truth labels to calculate the accuracy. The number of repetitions is three only for Dataset B. The training set of $\mathcal{D}_{T'}$ and test set are identical in Dataset A experiment.

## E   TWO DIMENSIONAL CO-VARIATE SHIFTS OBSERVATION

In this section, we confirm whether or not four datasets are following that two dimensional co-variate shifts assumption we introduced in the 2.2. We need to check dataset-wise, (1) existence of marginal distributions shifts between $\mathcal{D}_S, \mathcal{D}_T, \mathcal{D}_{T'}$ (2) mostly sharing of conditional distribution $P(y|x)$ between any domains. Apparently Dataset A follows that since the three data sets retain co-variate misalignment based on the semi-clockwise rotation action, and without this misalignment the labelling rules are perfectly consistent (Figure 8), likewise $\mathcal{D}_S, \mathcal{D}_T$ in Dataset B can be understood on the action of coloring digit part and background part (Figure 9). About the $\mathcal{D}_{T'}$ in Dataset B, although the explicit action does not return to $\mathcal{D}_S, \mathcal{D}_T$, and includes shifts about labelling rules as well as simple two dimensional co-variate shifts, we speculate a certain sharing of rule based on the facts that we can identify numbers with the human eyes.

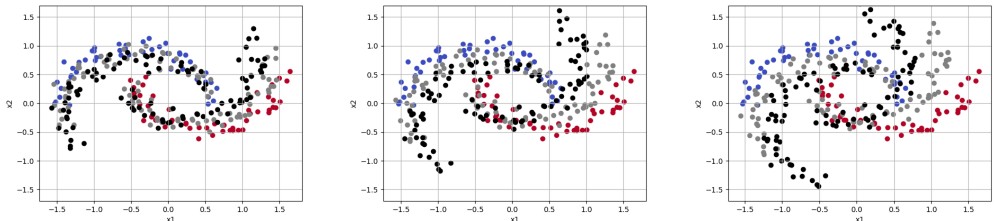

Figure 8: Data examples from Dataset A. Left is the source→15rotated→30rotated pattern, middle is 50 rotated target, right is 70 rotated target.

Previous work with Dataset C showed their co-variate shift existence between users, so they did the UDA experiment (Wilson et al., 2020) (e.g. see differences between user f and user g in Figure 10). It has been suggested in similar studies (Lane et al., 2011; Weiss & Lockhart, 2012) as well that the characteristics of the data measured will change depending on the age and user of the same behaviour, e.g. ML models trained on data from one age group will not generalise to different age groups. We additionally assume that different models of accelerometer equipment generate further shift based on the analysis by (Stisen et al., 2015) (e.g. see differences between model s3mini and model s3 in Figure 10). The measurement of the same phenomenon under static conditions with different models demonstrated that the distribution of measured values can be very different (please check Figure 1 in (Stisen et al., 2015)). Because all the data for a given model and a given user are taking a predetermined pattern of action classes, we can speculate that $P(y|x)$ is also sharing basically.

It is described by qualitative analysis of the previous study as co-variate shifts that satisfies a certain degree of shared labelling rules in the Dataset D (Oshima et al., 2024). The dynamics include different frequency of electricity peaks for certain time intervals between households (differences in the number of family members) or between seasons, while the previously mentioned rule of being at home if the electricity consumption is large and absent if it is small is shared (please check Figure 2

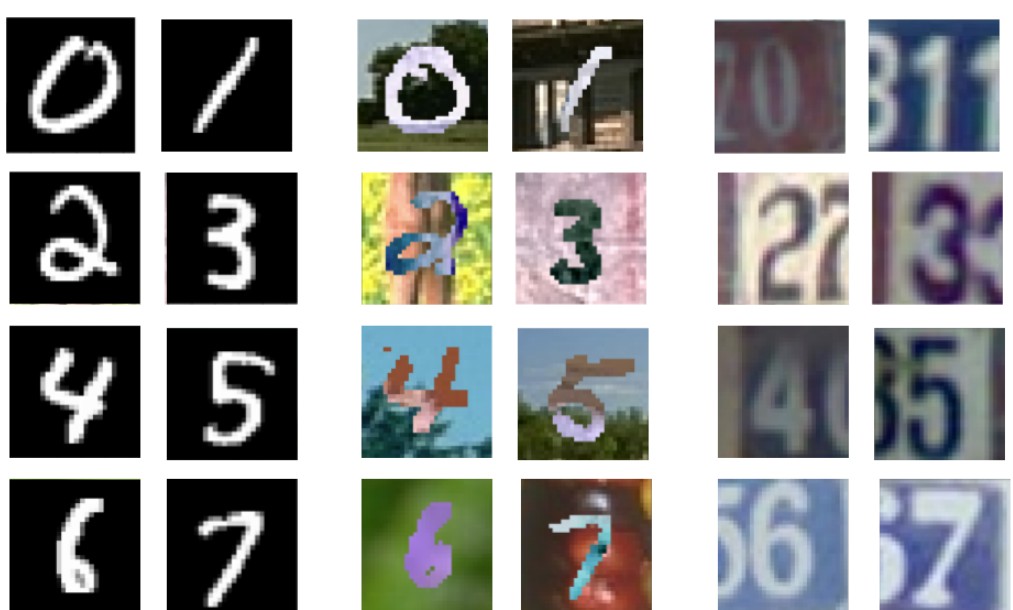

Figure 9: Data examples from Dataset B.

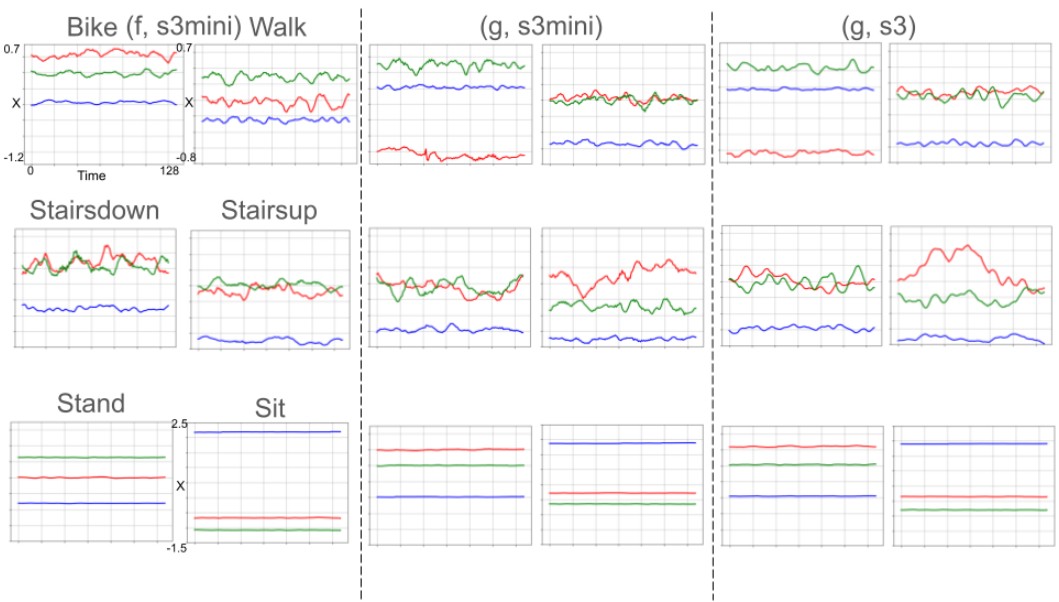

Figure 10: Data examples from Dataset C with (f,s3mini)→(g,s3). We omitted the plots of gyroscope data, blue plots correspond to the average of x-axis accelerometer data, red plots to the average of y-axis and green plots to the average of z-axis.

in (Oshima et al., 2024)). In particular, the Dataset D subjects live in Switzerland, where electricity consumption tends to vary significantly between summer and winter, and is generally higher in winter.

## F    APPLIED RESEARCHES' FUTURE RESEARCH DIRECTION

A promising direction for applied research is to evaluate the method with data and tasks that are in high demand for other social implementations, and to promote the application of UDA in business and the real world. For example, the problem of determining whether a patient with acute hypoxemic respiratory failure died during hospitalisation from medical data (e.g. blood pH and arterial blood oxygen partial pressure) and the problem of classifying the name of the disease,(Purushotham et al., 2017) may result in distribution shifts due to two dimensional data domains between different ages and different sexes. Semantic image segmentation in self-driving also may put a need for UDA between whether conditions and between a.m. or p.m. e.g. (Noon, Sunny)→(Noon, Rainy)→(Night, Rainy) (Liu et al., 2020).

## G    JDOT VERSION TWO STAGES DOMAIN INVARIANT LEARNERS AND EXPERIMENTAL VALIDATION

Algorithm and schematic diagram are in Algorithm 4 and Figure 11. In the pseudo code, we denote optimal transport solution for sample $i$ and sample $j$ as $OT_{i,j}$ for short. Figure 12 shows two-stages JDOT's superiority to previous studies, namely we found that "Upper bound > Ours > Max(Step-by-step, Normal, Lower bound)" for 4 out of 4 datasets. Figure 13 also says that evaluation for ours is always better than Normal and Step-by-step when any rotated target data in Dataset A.

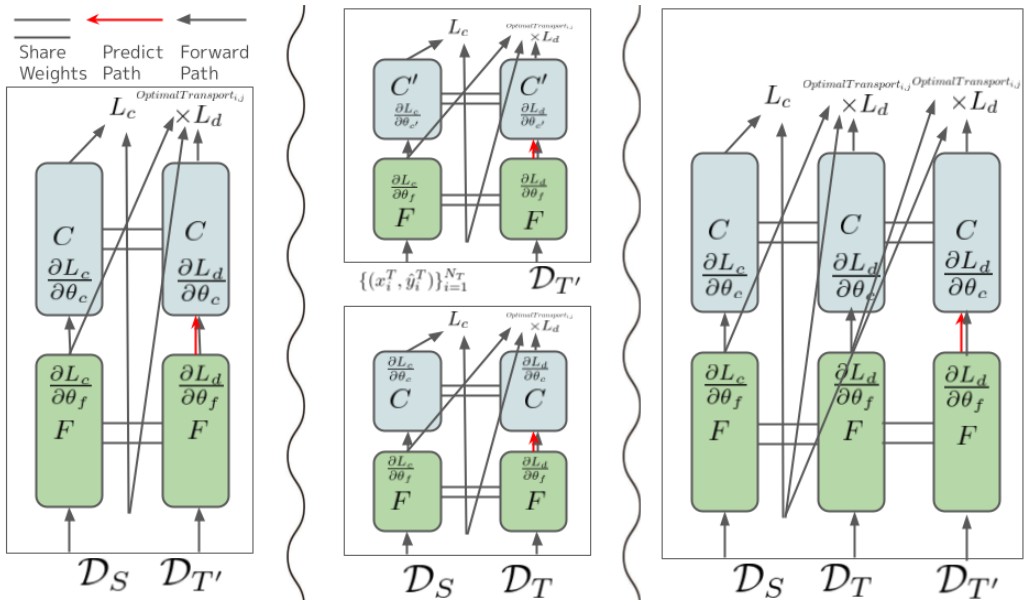

Figure 11: Forward path and backward process of Normal(JDOT), Step-by-step, Ours. The $L_{task}, L_{domain}$ as $L_c, L_d$ for short. The $OptimalTransport_{i,j}$ in this figure is optimal transport solution for source sample $i$ and target sample $j$.

**Algorithm 4** 2stages-JDOT

---

**Require:** source,intermediate domain,target $\mathcal{D}_S, \mathcal{D}_T, \mathcal{D}_{T'}$
**Ensure:** neural network parameters $\{\theta_f, \theta_c\}$

1: $\theta_f, \theta_c \leftarrow \text{init}()$
2: **while** epoch_training() **do**
3:     **while** batch_training() **do**
4:         $\hat{\mathbb{E}}(L_{domain}(\mathcal{D}_S, \mathcal{D}_T)) \leftarrow$
        $\frac{1}{batch \times batch} \sum_{i=1}^{batch} \sum_{j=1}^{batch} \{||F(x_i^S) - F(x_j^T)||^2 + CE(C(F(x_j^T)), y_i^S)\} \times OT_{i,j}$
5:         $\hat{\mathbb{E}}(L_{domain}(\mathcal{D}_T, \mathcal{D}_{T'})) \leftarrow$
        $\frac{1}{batch \times batch} \sum_{i=1}^{batch} \sum_{j=1}^{batch} \{||F(x_i^T) - F(x_j^{T'})||^2 + CE(C(F(x_j^{T'})), y_i^S)\} \times OT_{i,j}$
6:         $\hat{\mathbb{E}}(L_{task}) \leftarrow \frac{1}{batch} \sum_{i=1}^{batch} CE(C(F(x_i^S)), y_i^S)$
7:         $\theta_c \leftarrow \theta_c - \frac{\partial(\hat{\mathbb{E}}(L_{task}) + \hat{\mathbb{E}}(L_{domain}(\mathcal{D}_S, \mathcal{D}_T)) + \hat{\mathbb{E}}(L_{domain}(\mathcal{D}_T, \mathcal{D}_{T'})))}{\partial \theta_c}$
8:         $\theta_f \leftarrow \theta_f - \frac{\partial(\hat{\mathbb{E}}(L_{task}) + \hat{\mathbb{E}}(L_{domain}(\mathcal{D}_S, \mathcal{D}_T)) + \hat{\mathbb{E}}(L_{domain}(\mathcal{D}_T, \mathcal{D}_{T'})))}{\partial \theta_f}$
9:     **end while**
10: **end while**

---

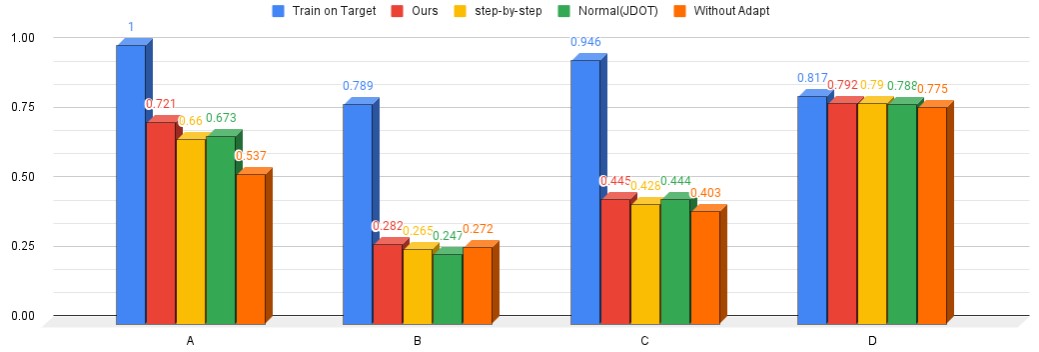

Figure 12: Quantitative result overview with JDOT

## H EMPIRICAL STUDY: EFFECTS OF TWO STAGES DOMAIN INVARIANT LEARNERS FREE PARAMETER INDICATOR

We analyzed whether there is a positive correlation between RV-based indicators (left of Theorem 3.1) and actual losses (1st term, right of Theorem 3.1). Using dataset A, learning rate was varied from $0.00000001 - 0.1$, and the RV-based score (cross entropy loss) was calculated for the model as the result of UDA learning. At the same time, actual losses (cross entropy loss) were calculated using target data for testing. Finally, pearson correlation coefficient was calculated to determine if there was a positive correlation between the two scores. Table 13 shows that there is a high positive correlation for all patterns. This result supports that two stages domain invariant learners free parameter indicator is useful in UDA experiments.

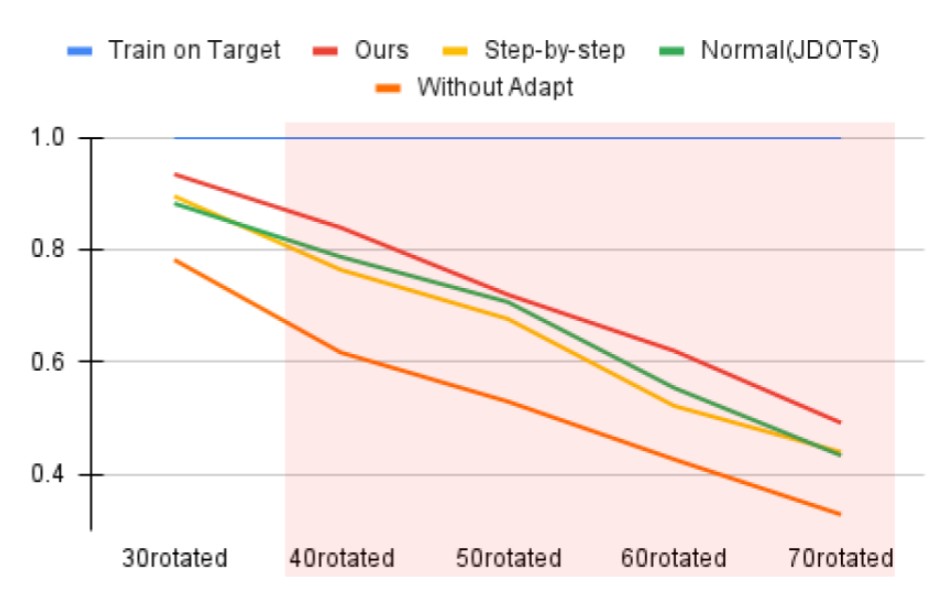

Figure 13: Quantitative result with JDOT in Dataset A

Table 13: Correlation measurements between Theorem 3.1 indicators and actual target ground truth losses using Dataset A and DANNs module.

| PAT | 15rotated→30rotated | | 20rotated→40rotated | | 25rotated→50rotated | | 30rotated→60rotated | | 35rotated→70rotated | |
|---|---|---|---|---|---|---|---|---|---|---|
| Method | RV based loss | Ground truth loss | RV based loss | Ground truth loss | RV based loss | Ground truth loss | RV based loss | Ground truth loss | RV based loss | Ground truth loss |
| 0.00000001 | 0.705 | 0.709 | 0.716 | 0.694 | 0.700 | 0.700 | 0.712 | 0.696 | 0.687 | 0.694 |
| 0.0000001 | 0.694 | 0.706 | 0.691 | 0.701 | 0.681 | 0.698 | 0.693 | 0.698 | 0.690 | 0.698 |
| 0.000001 | 0.685 | 0.698 | 0.730 | 0.699 | 0.709 | 0.700 | 0.690 | 0.657 | 0.690 | 0.686 |
| 0.00001 | 0.692 | 0.683 | 0.706 | 0.612 | 0.687 | 0.664 | 0.713 | 0.950 | 0.693 | 0.689 |
| 0.0001 | 0.713 | 0.547 | 0.707 | 0.785 | 0.720 | 1.47 | 0.713 | 0.950 | 0.704 | 1.72 |
| 0.001 | 0.683 | 0.121 | 0.729 | 0.683 | 0.691 | 0.550 | 0.686 | 0.333 | 0.696 | 1.19 |
| 0.01 | 6.83 | 0.953 | 3.87 | 1.64 | 6.87 | 0.97 | 6.49 | 5.54 | 3.91 | 2.61 |
| 0.1(learning rate) | 14.7 | 11.7 | 16.1 | 12.6 | 22.3 | 33.5 | 3.65 | 13.4 | 5.05 | 11.8 |
| Corr | | 0.92 | | 0.992 | | 0.960 | | 0.671 | | 0.859 |

# I   BENCHMARKS DESCRIPTION FOR 4 EXPERIMENTAL VALIDATION

We adopted six benchmark models for the comparison test and described input when training, input when inference and what the meaning is when compared to Ours.

Table 14: Benchmarks list.

| Method | Input When Training | Input When Inference | Role and Note |
|---|---|---|---|
| Train on Target | Training data of $\mathcal{D}_{T'}$ with its labels | Test data of $\mathcal{D}_{T'}$ | Call as Upper bound. Training $F$, $C$ with target ground truth labels (other methods cannot access to), validation on its test data. |
| Normal(DANNs,CoRALs) | $\mathcal{D}_S$, training set of $\mathcal{D}_{T'}$ | same as above | Normal domain invariant learners. Our methods should be better than these. Internal layers are from (Ganin et al., 2017; Wilson et al., 2020) (Oshima et al., 2024) . |
| Step-by-step | identical to ours $\mathcal{D}_S$, $\mathcal{D}_T$ and training data of $\mathcal{D}_{T'}$ | same as above | Step by step domain invariant learners. Our methods should be better than these. We can implement this with CoRALs easily. (Oshima et al., 2024) did not do that. |
| Without Adapt | $\mathcal{D}_S$ | same as above | Call as Lower bound. Ordinary supervised learning with source then validated on target test data. |

