# OpenReview forum: "TWO STAGES DOMAIN INVARIANT REPRESENTATION LEARNERS SOLVE THE LARGE CO-VARIATE SHIFT IN UNSUPERVISED DOMAIN ADAPTATION WITH TWO DIMENSIONAL DATA DOMAINS"
_ICLR.cc/2025/Conference — ICLR 2025 Conference Withdrawn Submission_

### Official Review · Reviewer_kyvG · 2024-10-24

**Soundness:** 2
**Presentation:** 1
**Contribution:** 1
**Rating:** 3
**Confidence:** 4

**Summary:**

This paper addresses large co-variate shift problems in UDA, particularly when dealing with two-dimensional co-variate shifts. The proposed method uses an intermediate unsupervised dataset to bridge the gap between source and target domains, learning domain-invariant features simultaneously between source-intermediate and intermediate-target pairs, which helps achieve better domain adaptation compared to direct source-target adaptation. The authors also derive a theorem for measuring the gap between trained models and unsupervised target labelling rules, which helps optimize free parameters without access to target labels. The proposed method is validated on classification 4 datasets including 38 UDA tasks.

**Strengths:**

1. The paper introduces a novel and effective solution to a practical challenge in machine learning by using intermediate data to bridge large domain gaps.
2. The proposed method is versatile as it can be integrated with various domain invariant representation learning techniques.
3. The authors derive a theorem for measuring the gap between trained models and unsupervised target labelling rules for hyper-parameters searching.

**Weaknesses:**

1.	This paper requires substantial improvement in terms of writing quality and clarity of presentation. (1) Many notations are without proper definition, (e.g., D_S, \hat{y}^S_{i,j}, N appear before being formally introduced); (2) The language is frequently imprecise and contains awkward expressions that hinder understanding. I strongly recommend the authors to thoroughly revise the mathematical notations with clear definitions and improve sentence structures and word choices for greater presentation.
2.	The authors assume the existence and availability of appropriate intermediate domains that perfectly fit their "two-dimensional domain shift" framework, but do not adequately address how to identify or construct such intermediate domains in real-world applications. For example, while it is intuitive to have MNIST-M as an intermediate domain between MNIST and SVHN, in most real-world scenarios, it is unclear: (1) How to systematically identify the two dimensions of domain shift; (2) How to obtain or construct suitable intermediate domain data; (3) What to do when clean intermediate domains are unavailable or when domain shifts are more complex than two-dimensional. Without addressing these practical concerns, the proposed method may have limited applicability in real-world domain adaptation problems.
3.	The technical novelty of the proposed method is limited. The approach is essentially a straightforward extension of existing domain-invariant learning methods. It merely splits the domain adaptation loss into two components (L_domain(S,T) + L_domain(T,T')) and applying standard adversarial training techniques, which lacks novel methodological designs in terms of loss function formulation, optimization strategy, or network architecture.
4.	The proposed parameter selection method is incremental. The proposed approach is essentially a straightforward adaptation of the existing Reverse Validation (RV) method to the two-stage domain adaptation setting, without substantial methodological innovations.
5.	The experimental evaluation of the paper is not up to current standards in domain adaptation research. The comparisons are limited to classical UDA methods like DANNs and Deep CORAL, while ignoring numerous recent advanced techniques that have shown significant improvements in handling large domain gaps.

**Questions:**

Please refer to weaknesses.

---

> ### Author Response · Authors · 2024-11-23
> **1.1. Notation errors**
>
> We authors worked together to improve notation and language issues (we are going to revise remaining ones before the final submission).
> Here is the notation revisions list.
>
> |(Page, Section, Paragraph) or Equation No. or Algorithm No.|Before|After|
> |---|---|---|
> |(2, 2.1, 1)|`\mathcal{D}_S w/o explanation`|`added "(Source domain, intermediate domain, target domain respectively)" at 1st position of \mathcal{D}_S`|
> |(3, 3.1, 1)|`\hat{y}^S_{i,j} w/o definition`|`revised as "We define $L_{task} = -\sum_{i=1}^{batch\_size} \sum_{j=1}^{num\_class} y_{i,j}^S\log(\hat{y}_{i,j}^S)$ as the cross entropy loss with a predicted probability $\hat{y}_{i,j}^S$ for source input.".`|
> |(3, 3.1, 1)| `N w/o definition` | `same as above(replace N with batch\_size).` |
> |(2, 2.1, 1)|`N_S, N_T, N_{T'} w/o definition`|`revised as "$N_S, N_T, N_{T'}$ as the sample sizes for source, intermediate and target.".`|
> |(3, 3.1, 1)|`"L_{domain}" w/o explanation`|`added "\textcolr{orange}{(measurement of distribution gap between source and target, we elaborate later)}"`|
> |(2, 2.1, 1)|`P_T(Y\|X), P_{T'}(Y\|X)`|`revised as "Then let $P_S(y\|x), P_S(x)$ denote the marginal and conditional distribution of source\textcolor{orange}{, defined for intermediate and target similarly}."`|
> |(3, 3.1, 2)|`"$\hat{\mathcal{D}}_S, \hat{\mathcal{D}}_{T'}$ with size $N$ from $\mathcal{D}_S, \mathcal{D}_{T'}$" w/o appropriate explanation`|`revised as “\textcolor{orange}{$\hat{\mathcal{D}}_S, \hat{\mathcal{D}}_{T'}$ be empirical sample sets with size $N$ drawn from source joint distribution (features and label) and target marginal distribution (features)}”`|
> |(3, 3.1, 2)|`"$d_i$ as the domain label" w/o appropriate explanation `|`revised as "$d_i$ as the domain label \textcolor{orange}{(binary label identifying source or target, $d_i \in \{0, 1\}$)}"`|
> |Equation(3), (4)|`"\mathbb{E}_{T'}(h), \hat{\mathbb{E}}_S(h), \mathbb{E}_{S}(h)" w/o definition`|`added "\textcolor{orange}{$\mathbb{E}_{T'}(h), \mathbb{E}_{S}(h)$ be expectations for hypothesis $h$ (task classification), $\hat{\mathbb{E}}_S(h)$ be the empirical expectation}".`|
> |Equation(5)|`"\mathbb{I}[h(x_i^S) \neq d_i^S]" w/o definition`|`added "\textcolor{orange}{$\mathbb{I}[\cdot]$ be the function outputting 1 when true inside the square brackets otherwise outputting 0}"`|
> |Algorithm 1 and Algorithm 3 and Algotithm 4|`d(partial differentiation notation) w/ difficulty in understanding the difference from domain label d_i.`|`replaced them as \partial`|
> |(6, 4.1, 1)|`n w/o definition`|`added "\textcolor{orange}{$n$ as the sample size of target data for testing}".`|
> |(6, 4.1, 1)|`"$\mathrm{accuracy}(x^{T'})=\frac{1}{n}\sum_{x^{T'}}[C(F(x^{T'}))=y^{T'}]$", lack of \mathbb{I}[]`|`added "\textcolor{orange}{\mathbb{I}[C(F(x^{T'}))=y^{T'}]".`|

---

> > ### Author Response · Authors · 2024-11-23
> > **1.2. Language errors**
> >
> > Here is the language revisions list.
> >
> > |(Page, Section, Paragraph)|Before|After|
> > |---|---|---|
> > |(1, 1, 2)|"did not be over" |"was not over" |
> > |(10, 4.3, 3)|"It can be read that it is difficult to classify these human's behaviour classes" (not direct expression)|"It is difficult for them to classify these human's behaviour classes"|
> > |(2, 2.2, 1)|"In the stree or not"|"In the street or not" |
> > |(2, 2.1, 1)|"They are in the homogeneous domain adaptation assumption, that is why they share same sample space but different distribution." (bad sentence connection, is not cause-and-effect)|"They are in the homogeneous domain adaptation assumption, namely they share ... "|
> > |(5, 3.1, 5)| "usecases"| "use cases"|
> > |(3, 2.2, 1)| "We investigated to what extent they follow that assumption dataset-wise in the Appendix E." (not direct expression) | "We investigated whether or not each dataset follows this assumption in the Appendix E." |
> > |(3, 3.1, 2)|"bounded by distribution difference between source and target" (w/o explanation about which distribution?)|  "bounded by marginal distribution ..."  |
> > |(4, 3.1, 3)| "Each of these has differences in the construction of $L_{domain}$, with DANNs building it by a classification problem between domains, CoRALs building it by the distance between the covariance matrices of $C$, DANs building it by the multiple kernel variant of maximum mean discrepancy (MK-MMD \citep{gssbpfs}) calculated on $F$ and $C$, and the Deep Joint Distribution Optimal Transportation (DeepJDOT) uses wasserstein distance based on the optimal transport problem to measure $L_{domain}$ \citep{dkftc}."  (bad sentence connection) |"The construction of $L_{domain}$ differs across methods. DANNs define it as a loss of domain classification problem, while CoRALs use the distance between covariance matrices of $C$. DANs build it using the multiple kernel variant of maximum mean discrepancy (MK-MMD \citep{gssbpfs}) calculated on $F$ and $C$, and Deep Joint Distribution Optimal Transportation (DeepJDOT) define it by wasserstein distance based on the optimal transport problem \citep{dkftc}."|
> > |(6, 4.1, 1)|"The all codes we used in this paper is available on GitHub"|"codes we used in this paper are available ..."   |
> > |(6, 4.1, 2)|"MNIST-M whose background and digit itself copied from MNIST is replaced by colored one"  (not direct expression)|  "MNIST-M which is the randomly colored version of MNIST"|
> > |(7, 4.2, 2) |   "The Figure 3 shows our methods' superiority to previous studies in Dataset A-D(with DANNs), D(with CoRALs), thus the ideal state ... experiment." (bad sentence connection, is not cause-and-effect) |  "The Figure 3 shows our methods' superiority ... , namely the ideal state ... experiment."|
> > |(8, 4.3, 1)| "It can also be seen that the task classifiability to the source is simultaneously optimised and eventually asymptotically approaches zero or small value." (not direct expression)  | "Also we found that task classifiability ... "   |
> > |(10, 4.3, 2)|"representation at feature level are colored in two ways"|"representation at feture level is colored .."  |
> > |(4, 3.1, 3) |”One should take into account that …”|”Please note that …”|

---

> ### Author Response · Authors · 2024-11-23
> **2. Concerns about the intermediate domain**
>
> # (1)How to identify two attributes in datasets
> Systematically, differences in factors (environment, time, user, style) when measuring data define two attributes. Databases normally have such information in the companies using the energy or accelerometer or camera sensor so automatic detection is easy.
> # (2)How to obtain suitable intermediate domain
> For time series data, databases in the companies using the energy or accelerometer sensor normally hold big data in this digital age. So it is too rare for databases not to have one specific intermediate domain(i.e. needed for our method) frequently. For other data types including image data, you can't always get a suitable intermediate, this should be a future research question e.g. generative models possibly will create suitable one based on prompting.
> # (3)solution to higher dimensional domain shift
> Our method is scalable to higher dimensional domain shift technically by adding domain invariant learners in parallel. E.g. in a three dimensional case with (Domain1 AorB, Domain2 AorB, Domain3 AorB) , we need three domain invariant learners like below.
>
> 1st domain invariant learner tackles:  (Domain1 A, Domain2 A, Domain3 A) vs  (Domain1 A, Domain2 B, Domain3 A)
> 2nd domain invariant learner tackles:  (Domain1 A, Domain2 B, Domain3 A) vs (Domain1 A, Domain2 B, Domain3 B)
> 3rd domain invariant learner tackles: (Domain1 A, Domain2 B, Domain3 B) vs (Domain1 B, Domain2 B, Domain3 B)
>
> This also should be another possible future research direction.
> # Conclusion: Rebuttal to limitations in the real world UDA
> Exact applicability(current state of our paper) to real world UDA includes most situations with time series data, limited situations with other data types. Based on our references (in the paper) and knowledges three dimensional or higher domains UDA with time series data is not explored in the real world projects or researches which aim for applications, because we can imagine that but it’s too rare for an additional domain to cause a certain amount of distribution shift in the existing data.
>
> We conclude that our contribution in applicability is fair.

---

> ### Author Response · Authors · 2024-11-23
> **3. and 4. Concerns about novelty of two stage domain invariant learners and RV based free parameters optimization**
>
> We argue again that our straightforward extension (algorithms, theorem and its proof)  is powerful and scalable to future UDA works. We have confidence that researchers in this field will get enough knowledge from our paper.

---

> ### Author Response · Authors · 2024-11-23
> **5. Concern about benchmarks**
>
> First we argue that the recent trends (we talk about prompt based, CLIP based and Vision Transformer based) in UDA are limited in the data type. We cannot adopt these methods for time series data (Dataset C, D in our paper).
>
> Second, we added another newer benchmark (DeepJDOT from ECCV2018 with any data type) to our revision paper, and we found consistent results. Here is the main result and validation with Dataset A is in Figure 13 in the revision paper.
> ||Train on Target|Ours|Step-by-step|JDOT|Without Adapt|
> |---|---|---|---|---|---|
> |A|1|__0.721__|0.66|0.673|0.537|
> |B|0.789|__0.282__|0.265|0.247|0.272|
> |C|0.946|__0.452__|0.427|0.443|0.403|
> |D|0.817|__0.792__|0.79|0.788|0.775|

---

### Official Review · Reviewer_pMKB · 2024-10-27

**Soundness:** 2
**Presentation:** 1
**Contribution:** 1
**Rating:** 1
**Confidence:** 3

**Summary:**

The authors of this manuscript propose a two-stage domain-invariant representation learning method, which uses semantic intermediate data to bridge the gap between source and target domains. This method improves classification performance even under large covariate shifts by learning domain-invariant features and optimizing task discriminability through source labels. The paper also introduces a theorem for optimizing free parameters by measuring the gap between trained models and target labeling rules. The proposed method outperforms previous UDA techniques across 38 tasks in 4 representative ML datasets.

**Strengths:**

1. A new approach was proposed
2. It provides an automated free-parameter tuning method without needing access to target ground truth labels.

**Weaknesses:**

1. English was not used properly:
    a) line 52: "did not be"
    b) line 498: "It can be read that" sounds awkward.
    ...
Lots of sentences in this manuscript are not authentic, making it hard to follow the manuscript's content. I strongly recommend authors taking times to improve the presentation of this work.

2. No related works section.

3. The x, y axis labels for figure 4-6 are not easily visible.

**Questions:**

Use cases of the proposed methods are too limited. It was designed for two dimensional data. What if the source and target data are in high dimensional space?

---

> ### Author Response · Authors · 2024-11-23
> **Misunderstanding about “It was designed for two dimensional data.”**
>
> __It was not designed for two dimensional data, works for high dimensional data (e.g. image, time series data).__
> __Please read the revision paper again based on this discussion?__
>
> (p.s.)
> We revised the language issues.
> Related works are described in section 3.1 and  section 5, also we added the latest UDA papers including the papers you proposed to revision papers.

---

### Official Review · Reviewer_fr2j · 2024-10-31

**Soundness:** 3
**Presentation:** 2
**Contribution:** 2
**Rating:** 3
**Confidence:** 4

**Summary:**

This paper proposes a simple two-stage domain adaptation method by feeding source domain, intermediate domain and target domain into the model. The intermediate domain overcomes the large covariate shift problem that is widely a challenge of domain adaptation. This paper further proposes a free parameter indicator with reverse validation strategy.


Based on the authors' feedback and other reviewers' comments, I would reduce my final rating on this work.

**Strengths:**

1. The two-stage domain adaptation method, i.e. two-stage DANN sound good and interesting.
2. The reverse validation based idea is interesting and not frequent in domain adaptation community.
3. The two-stage strategy can be scalable to other method such as CORAL.

**Weaknesses:**

1. Domain adaptation has undergo a wide study in the past decade. However, with the multimodal large language model, domain gap has been allievated from another way, such as CLIP based [1,2]. Therefore, in this submission, the impact of this proposal may be weak.
2. There lacks sufficient comparisons to previous SOTAs in recent works [3], particularly large vision-language model based and prompt based [4].
3. The writting should be further improved for easier reading.
4. Using intermediate domain as a bridge is not new because there are a wide research in DA with intermediate state [6, 7].
5. As the algorithms of 2 stage DANN shows, the intermediate domain is required. I thinks how to obtain intermediate domain is still an open question. This is not discussed.
6. Transformer based DA models should also be discussed since it has been widely used in domain adaptation [8, 9].

[1] Learning transferable visual models from natural language supervision. In ICML, pages 8748–8763. PMLR, 2021.
[2] Ad-clip: Adapting domains in prompt space using clip. In ICCV, pages 4355–4364, 2023.
[3] Gradient Harmonization in Unsupervised Domain Adaptation. In IEEE TPAMI, 2024.
[4] Domain adaptation via prompt learning. In IEEE TNNLS, 2023.
[5] Domain-Agnostic Mutual Prompting for Unsupervised Domain Adaptation. In CVPR, 2023.
[6] Semi-Supervised Domain Generalization with Evolving Intermediate Domain. In PR, 2023.
[7] Manifold Criterion Guided Transfer Learning via Intermediate Domain Generation. In IEEE TNNSL, 2019.
[8] Tvt: Transferable vision transformer for unsupervised domain adaptation. In WACV, 2023.
[9]  Safe self-refinement for transformer-based domain adaptation. In CVPR, 2022.

**Questions:**

1. If the reverse validation like idea is always effective?
2. This is more like a training strategy by commenting on step-by-step manner comparing to end-to-end manner.

---

> ### Author Response · Authors · 2024-11-23
> **Question2**
>
> Sorry we cannot understand what the question is, can you clarify it?

---

> ### Author Response · Authors · 2024-11-23
> **We start rebuttal based on the papers you proposed**
>
> We really enjoyed understanding the papers you suggested (we love them). We authors have worked together to understand most of the pages of all the papers. Here is a summary of our understanding, and we rebut based on this table.
>
> # Reference Table
> |Method Name|In Short Words|Difference from Ours|Orthogonal to Ours|
> |---|---|---|---|
> |CLIP[1]|Proposed CLIP itself| | |
> |Ad-clip[2]|Prompt based UDA and use CLIP as backbone, domain prompt is automatically learned|limited to computer vision (CV) task technically|Yes|
> |GH[3]|Mathematical control for two gradients(based on task classification loss and domain align loss) improved UDA. |limited to computer vision (CV), experimentally(unclear scalable to time series data)|Yes|
> |DAPL[4]|Prompt based UDA, domain prompt is manually created|same as [2]|Yes|
> |DAMP[5]|prompt based UDA and large vision-language model backbone improves performance|same as [2]|Yes|
> |APL and DCG[6]|Generated intermediate domain improves UDA|same as [2](explain in detail later)|Basically No|
> |MCTL[7]|Manifold Criterion instead of MMD generates intermediate domain and improves UDA|same as [3]|Basically No|
> |Tvt[8]|vision transformer backbone improves UDA| same as [2]|Yes|
> |SSRT[9]|vision transformer backbone and consistency regularization improve UDA|same as [2]|Yes|

---

> ### Author Response · Authors · 2024-11-23
> **Weakness 1**
>
> Based on table, papers ([1][2]) are limited to CV tasks since papers depend on the strong assumption i.e. can explain content and style information for image in natural language. This is not applicable to time series data due to difficulty to explain time series data in natural language except in very simple cases e.g. where acceleration is not there at all and the person is just sitting down.
>
> Our method can be applicable to any data type.We argue that the impact of this proposal is not weak.
>
> On the other hand,  papers[1][2] can be used as an internal UDA module in our two-stages domain invariant learners to improve Dataset B. For the paper[2], By extending source-target domain tokens to source-target-intermediate domain tokens, the same optimization should work well (Additional L_{smn} loss is not in our equation (1), but this is not a big thing).
>
>  We described this in the future research direction section.
>
> # (Added @5 Conclusion and future research direction)
> Also large vision-language models' prompting was used for UDA. They format domain invariant and task variant information as the prompting and showed impressive performance improvements in image recognition datasets[2].

---

> ### Author Response · Authors · 2024-11-23
> **Weakness2**
>
> Based on the table, papers[3][4] are limited to CV as well.
> We added another newer benchmark (DeepJDOT from ECCV2018 scalable to  any data type) to our revision paper instead of [3][4], and we found consistent results. Here is the main result (compatible with Figure 3 in the revision paper) and validation with Dataset A is in Figure13 in the revision paper.
> ||Train on Target|Ours|Step-by-step|JDOT|Without Adapt|
> |---|---|---|---|---|---|
> |A|1|__0.721__|0.66|0.673|0.537|
> |B|0.789|__0.282__|0.265|0.247|0.272|
> |C|0.946|__0.452__|0.427|0.443|0.403|
> |D|0.817|__0.792__|0.79|0.788|0.775|
>
> We conclude that our comparison test does not lack sufficient comparisons (please note scope difference).

---

> ### Author Response · Authors · 2024-11-23
> **Weakness4**
>
> Based on papers[6][7], you are right so we revised sentences.
>
> Although Papers [6][7]  are limited to CV, paper [6] generates synthetic intermediate data via ax_i + (1-a)u_j (a: 0<=a<=1, x_i:labeled data, u_i: unlabeled data).  This generation is bad for time series data, since time dependence will be broken. Paper [7] based on a mathematical idea (i.e. focusing on non-linear local structure of data across domains might improve UDA) lacks its validity outside of CV.
>
> We think our additional sentences describing recent works in the revision paper is enough for this discussion.
>
> # original
> This two dimensional domains assumption and solution to large co-variate shift problem have not been explored so far (except for (Oshime et al., 2024)).
> # revision
> Two dimensional domains assumption is novel in this field, although uses of an intermediate domain to resolve large co-variate shift are explored in (Lin et al., 2021; Zhang et al., 2019; Oshima et al., 2024)as well. Their experiments showed a synthetic intermediate domain can improve UDA (Lin et al., 2021; Zhang et al., 2019), but limited to computer vision tasks technically or experimentally. Our proposition is not limited to a specific data type.

---

> ### Author Response · Authors · 2024-11-23
> **Weakness5**
>
> For time series data, databases in the companies using the energy or accelerometer sensor normally hold big data in this digital age. So it is too rare for databases not to have one specific intermediate domain(i.e. needed for our method) frequently. Two papers below are describing this similarly.
>
> [11]Active Learning for Multivariate Time Series Classification with Positive Unlabeled Data. In International Conference on Tools for Artificial Intelligence (ICTAI), pages 178-185, 2015
> https://ieeexplore.ieee.org/document/7372134
> [12]Time series feature learning with labeled and unlabeled data, Pattern Recognition(Journal), vol 89, pages 55-66, 2019
> https://www.sciencedirect.com/science/article/abs/pii/S0031320318304473
>
> For other data types including image data, you can't always get a suitable intermediate. One way is to use the Web with huge unlabeled data. Alto this should be a future research question e.g. generative models possibly will create suitable one based on prompting or papers[3][4] methods might help us.
>
> We added this discussion to the revision paper, and we conclude that this is enough.
>
> # added @Conclusion and future research direction
> Another research question is how to obtain an intermediate domain. For time series data, databases normally hold huge amounts of unsupervised data[11][12], so two dimensional co-variate shifts assumption is quite natural. For other data types such as image data, your database can't always hold a suitable intermediate. One way is to use the Web with huge unlabeled data. Alto this should be a future research direction e.g. generative models possibly will create suitable one based on prompting or papers [3][4] methods might help create intermediate synthetically.

---

> ### Author Response · Authors · 2024-11-23
> **Weakness6**
>
> Based on the table, Tvt[8] and SSRT[9] are limited to CV tasks.
> We can use Tvt[8] and SSRT[9] as internal UDA modules in Dataset B experiment, and might improve performance.
>
> We talked about them in the revision paper.
>
> # added @Conclusion and future research direction
> Safe Self-Refinement for Transformer-based domain adaptation uses vision transformer backbone and consistency regularization as well [9], might improve the performance (vision transformer based UDA was also analyzed in another paper and improved the performance [8]).

---

> ### Author Response · Authors · 2024-11-23
> **Question1**
>
> Technically the idea is not limited to any sense(data type, internal layer, internal domain invariant representation learning module).

---

### Official Review · Reviewer_Y4ir · 2024-11-02

**Soundness:** 2
**Presentation:** 2
**Contribution:** 2
**Rating:** 6
**Confidence:** 3

**Summary:**

This paper introduces a two-stage domain-invariant representation learning approach for UDA under large co-variate shifts. It uses intermediate unsupervised data to bridge the gap between source and target domains. Additionally, a theoretical framework is proposed for parameter tuning without requiring target labels.

**Strengths:**

The paper addresses a critical limitation in UDA by focusing on large co-variate shifts in two-dimensional data domains, which are common in real-world applications such as autonomous driving, human activity recognition.

**Weaknesses:**

1.  The main limitation of the proposed method is its dependency on an intermediate, unsupervised domain that is semantically related to both the source and target domains. This dataset may not always be available or feasible to collect, especially in real-world applications with limited resources.
2. The practical application of the proposed method is not fully demonstrated. The benefits remain largely theoretical, making it hard for readers to grasp its relevance without concrete examples of its impact on model performance.
3. Lacks a deep discussion that contextualizes how this approach builds upon or diverges from existing UDA methods. An analysis/experiment can be done to show how the proposed method addresses specific limitations in prior approaches, such as domain-adversarial training or correlation alignment techniques.

**Questions:**

1. How would the method perform if an intermediate, unsupervised domain were unavailable or of low quality? Are there alternative approaches, such as synthetic data generation or transfer learning, that could help mitigate this dependency? Could the authors provide guidance on selecting or creating intermediate datasets for cases where a semantically related domain is not readily available?

2. Can the authors provide specific examples or case studies to demonstrate the practical impact of the proposed parameter tuning framework on model performance? Would an experiment isolating the effects of the parameter tuning framework help clarify its practical benefits? If so, could the authors consider adding one to the paper?

3. How does the proposed method address specific limitations of existing UDA techniques, such as domain-adversarial training or correlation alignment? Could the authors provide a comparison experiment or detailed analysis that highlights the advantages and trade-offs of this two-stage approach relative to traditional UDA methods?

4. How does the proposed method handle highly heterogeneous domains where intermediate data is noisy or contains varying domain characteristics?

5. Could the authors include ablation studies to isolate the effects of each stage in the two-stage process? This might help clarify the contributions of each component in achieving domain invariance.

---

> ### Author Response · Authors · 2024-11-23
> **Weakness1**
>
> Yes, solutions to an unavailable intermediate domain should be a future research question, and there are a lot of ways e.g. generating intermediate domain based on prompting with large language models, using the Web with huge unlabeled databases and using synthetic intermediate domain generation algorithm(only for image data).
>
> We added this discussion to the revision paper.
> # added @Conclusion and future research direction
> Another research question is how to obtain an intermediate domain. For time series data, databases normally hold huge amounts of unsupervised data[11][12], so two dimensional co-variate shifts assumption is quite natural. For other data types such as image data, your database can't always hold a suitable intermediate. One way is to use the Web with huge unlabeled data. Alto this should be a future research direction e.g. generative models possibly will create suitable one based on prompting or papers [3][4] methods might help create intermediate synthetically.

---

> ### Author Response · Authors · 2024-11-23
> **Question 2**
>
> We added an experiment isolating the effects of the parameter tuning framework in the revision paper Appendix H. Results show that there is enough correlation between RV based scores and its goals (target data’s ground truth loss), we again argue that this indicator is useful.

---

### Official Review · Reviewer_2tRr · 2024-11-03

**Soundness:** 1
**Presentation:** 1
**Contribution:** 1
**Rating:** 1
**Confidence:** 5

**Summary:**

This paper proposes a two-stage domain invariant representation learning method to address large co-variate shifts in unsupervised domain adaptation. The approach uses intermediate, unlabeled data to create smoother transitions between source and target domains, aiming to enhance classification performance under challenging conditions. The authors claim their method outperforms existing UDA models, especially when co-variate shifts are significant.

**Strengths:**

The idea of utilizing intermediate data to smooth the domain adaptation process sounds reasonable.

**Weaknesses:**

1. Poor clarity and organization. The paper is challenging to read, with many grammatical errors, convoluted language and unclear explanations of the methodology.
2. Lack of rigorous validation: The theoretical claims, especially the effectiveness of two-stage learning and parameter optimization, lack sufficient mathematical justification and empirical support. In line 62, the authors claim that "intermediate data (unsupervised) between source and target to ensure simultaneous domain invariance between source and intermediate data and invariance between intermediate and final target data". Doesn't this imply the source and target data are domain invariant as well?
3. The text is verbose and repetitive, making it difficult to extract key insights and understand the novelty compared to prior methods.

**Questions:**

See weaknesses.

---

> ### Author Response · Authors · 2024-11-23
> **Misunderstanding aboud “Doesn't this imply the source and target data are domain invariant as well?”**
>
> __The answer is yes.__
>
> __Our paper and all other domain invariant learning papers (e.g. DANNs from JMLR, DANs from ICML2015, CoDATS from KDD2020 and AD-CLIP from ICCV2023) are basically doing this in a different way (so this is not strange thing).__
> __We proposed the decomposition of domain invariant loss using an intermediate domain since it will ease learning in the meaning of UDA when compared to other papers.__
>
> This leads to task(1) acquisition of domain invariance between source and intermediate domain, discriminability for intermediate domain (F, C, D in Figure1 ours) and task(2) acquisition of domain invariance between intermediate domain and target, discriminability for target (F, D’ in Figure1 ours, based on theory (equation(3)) discriminability for intermediate domain and intermediate(domain)-target(domain) domain invariance will lead to discriminability for target).
>
> Our hypothesis is that each task should be easier than normal (equation(1)) since the gap between source and target is larger than the gap between two data domains in task(1) or task(2), thus overall our method will ease learning convergence in the meaning of UDA.
>
> Experimental validation including 4 datasets (image recognition, time series classification, table data classification) and 38 UDA tasks supported our hypothesis.
>
> __Please read the revision paper again based on this discussion?__
>
> (p.s.)
> We revised language issues.

---

### Author Response · Authors · 2024-11-23
**We appreciated everyone for review process.**

We deeply appreciated everyone for all of review processes, and added the revision paper.

|textcolor|role|
|---|---|
|orange|revised notation issues without changing meaning|
|green|revised language issues without changing meaning|
|blue|revised or added content itself including additonal benchmarks and survey of the latest UDA woks mainly in computer vision|

---

### Note · Authors · 2024-11-28

**Comment:**

I have read and agree with the venue's withdrawal policy on behalf of myself and my co-authors.

Thank you a lot for review.
We've decided to withdraw this paper.

**Withdrawal Confirmation:**

I have read and agree with the venue's withdrawal policy on behalf of myself and my co-authors.